# Seed-borne bacteria drive wheat rhizosphere microbiome assembly via niche partitioning and facilitation

Daniel Garrido-Sanz ✉ & Christoph Keel ✉

Microbial communities play a crucial role in supporting plant health and productivity. Reproducible, natural plant-associated microbiomes can help disentangle microbial dynamics across time and space. Here, using a sequential propagation strategy, we generated a complex and reproducible wheat rhizosphere microbiome (RhizCom) to study successional dynamics and interactions between the soil and heritable seed-borne rhizosphere microbiomes (SbRB) in a microcosm. Using 16S rRNA sequencing and genome-resolved shotgun metagenomics, we find that SbRB surpassed native soil microbes as the dominant rhizosphere-associated microbiome source. SbRB genomes were enriched in host-associated traits including degradation of key saccharide (niche partitioning) and cross-feeding interactions that supported partner strains (niche facilitation). In vitro co-culture experiments confirmed that helper SbRB strains facilitated the growth of partner bacteria on disaccharides as sole carbon source. These results reveal the importance of seed microbiota dynamics in microbial succession and community assembly, which could inform strategies for crop microbiome manipulation.

Plant-associated microbiomes carry a wealth of functions that are intrinsically related to the growth and health of the host[1–4] and collectively contribute to the functioning of the plant holobiont[5]. In agroecosystems, the rhizosphere microbiome (that is, rhizobiome) plays a critical role in crop productivity[6,7], and approaches to manipulate it hold great promise for meeting increasing agricultural demands without resorting to harmful chemical fertilizers and pesticides.

Targeted microbiome manipulation focuses on the addition, removal or modulation of specific functions within the rhizobiome, achieved through the introduction of plant-beneficial inoculants or the elimination of harmful pathogens[1–3,8–10]. However, the success of plant microbiome engineering is often constrained by a limited understanding of how complex communities respond to such interventions[11,12]. This is not surprising, as soil and rhizosphere microbiomes are among the most complex environments on Earth[13,14], making them particularly challenging for mechanistic studies. Bottom-up approaches using synthetic communities (SynComs) offer highly tractable and reproducible systems in which complex interaction mechanisms can be elucidated[15–17], yet they often represent an oversimplification of natural environments. In contrast, top-down strategies consist of the generation of natural communities (NatComs) by culturing microbiomes in controlled set-ups that mimic natural environments[18–20], allowing the selection of complex microbiomes that closely resemble the structure and composition of their original environment. Despite their potential, the widespread adoption of NatComs remains limited due to their inherent biological complexity, the lack of reproducible natural microbial communities, and the limitations of current technological and analytical methods.

The selection exerted by root exudates is arguably one of the main factors determining the rhizobiome assembly. Root exudates are composed of carbon-rich molecules, which support net bacterial growth in high numbers[21,22], and also signalling molecules that control the proliferation of specific bacterial taxa[23–25]. Owing to the diffusion of root exudates into the adjacent soil, soil-dwelling bacteria have long

Department of Fundamental Microbiology, University of Lausanne, Lausanne, Switzerland. ✉e-mail: daniel.garridosanz@unil.ch; christoph.keel@unil.ch

been considered the predominant source of rhizosphere microbes. However, recent studies have emphasized that seed-transmitted bacteria are also an important source of the rhizobiome[26–28]. These heritable bacteria are found across agriculturally relevant plants[29]. Bacteria reside in the internal tissues of seeds, which they reach by different transmission routes[26,30]. Upon germination, seed bacteria may leave the seed tissues and become part of the new plant rhizobiome, that is, seed-borne rhizosphere bacteria (SbRB). These bacteria contribute to beneficial functions for the plant[31–34], especially during early stages of plant development[35], independent of their availability in the surrounding soil. However, the mechanisms by which SbRB and soil microbiomes coalesce into the plant rhizobiome, and whether specific metabolic signatures drive their integration and function remain poorly investigated, hindering our ability to predict or manipulate these interactions.

In this study, we developed a sequential wheat rhizobiome propagation approach to obtain a stable and reproducible species-rich rhizosphere natural community (RhizCom) resulting from the coalescence of two microbiome sources: the soil microbiome and heritable SbRB transmitted via seeds. Plants were grown from surface-disinfected seeds in microcosms with sterilized soil matrix. This allowed us to investigate the successional microbiome dynamics leading to the assembly of a reproducible wheat rhizobiome constrained by plant root selection. Functional analyses revealed the enrichment of plant-interaction traits in the root-associated communities, as opposed to the broader metabolic diversity observed in the soil community. The specific metabolism of saccharides in SbRB demonstrates their critical role in facilitating rhizobiome assembly. This work provides an unprecedented view of early rhizobiome assembly in the world's third most productive crop, facilitating the acquisition of reproducible and complex bacterial communities aimed for future mechanistic studies.

## Results

### Sequential succession drives phylum-specific selection
To obtain a tractable wheat rhizobiome, we performed six successive cycles of microbiome propagation in the rhizosphere of wheat (*Triticum aestivum* cv. Arina) using a soil microbiome cell suspension as the initial inoculum. After 7 days of plant development, the rhizosphere was collected, and a rhizosphere cell suspension was obtained and used to re-inoculate the next cycle (Fig. 1a and Extended Data Fig. 1). The SbRB community was obtained from 1 cycle of 16 uninoculated plants. We tracked bacterial community dynamics using analysis of the small ribosomal subunit (16S) rRNA gene. A strong shift from the initial soil microbiome composition was observed, caused by the selection of bacteria mainly belonging to the phyla Proteobacteria and Bacteroidota, and to a lesser extent some members of Firmicutes and Verrucomicrobiota (Fig. 1b and Supplementary Fig. 1). SbRB and soil samples were the most dissimilar compared with the microbiome composition observed along the succession cycles, which showed a converging trajectory (Fig. 1c,d). Taxa selected in the initial cycles were maintained throughout the remaining steps (Fig. 1e and Supplementary Fig. 2), consistent with a stabilization of phylogenetic diversity (Spearman correlation, $R = −0.15$, $P = 0.49$) and colony-forming unit (c.f.u.) counts (Fig. 1f,g and Supplementary Table 1). However, a progressive decrease in Shannon diversity (Spearman correlation, $R = −0.85$, $P = 1.2 \times 10^{-7}$, Fig. 1h) and other alpha diversity indices was observed (Supplementary Fig. 3). These data demonstrate the establishment of a stable and diverse wheat rhizosphere community (RhizCom) under the microcosm conditions used, consisting of 464 amplicon sequence variants (ASVs) belonging to 126 distinct genera (Supplementary Table 2). The reproducibility of the RhizCom was validated by re-growing the cryopreserved RhizCom at −80 °C in the wheat rhizosphere, which had a similar taxonomic composition, number of ASVs and Shannon diversity as the community of the last propagation cycle (Fig. 1, and Supplementary Fig. 3 and Supplementary Table 3).

### Heritable SbRB constitute the major rhizobiome source
We examined the origin of RhizCom members by identifying exact matches of ASVs from either the SbRB or soil communities, as these are the only possible sources. Only a small number of RhizCom ASVs could be traced to either source community (11.4%, Fig. 2a and Supplementary Table 3). Specifically, we detected 33 ASVs originating from the soil inoculum and 25 from the SbRB in the RhizCom microbiome, which together accounted for 51% of the total RhizCom relative abundance. Notably, one *Flavobacterium* ASV and one *Serratia* ASV, both detected in the soil inoculum and the SbRB community, accounted for 43.3% of the RhizCom (Supplementary Table 3). Soil wash and SbRB ASVs contributing to the RhizCom collectively represented 4% and 84.7% of the relative abundance of their respective microbiomes. Most ASVs not traced to the source communities emerged throughout the succession cycles (70.9% of RhizCom ASVs), indicating that individual heritable SbRB heterogeneity was retained within the RhizCom (Supplementary Table 3). This was further evidenced by a decrease in ASV emergence over the cycles, mostly affecting low-abundance taxa (Fig. 2b). RhizCom ASVs not detected in any condition (82 ASVs, Fig. 2a) had a low representation in the RhizCom (85% of them below 0.01% of relative abundance, Supplementary Table 3) and could be attributed to specific SbRB heterogeneity within its cycle. These results demonstrate that SbRB dominate the rhizosphere assembly during early niche development, which can be attributed to a pioneering effect as they are the first to arrive in the developing rhizosphere environment.

The comparison between the initial soil inoculum and the RhizCom revealed the most striking changes, with 808 ASVs enriched in the soil wash versus 133 in the RhizCom (Fig. 2c–f), followed by SbRB versus RhizCom (14 versus 313 ASVs, respectively). This indicates substantial differences in community composition, suggesting a strong host-mediated selective pressure. The enrichment of ASVs in the RhizCom mainly affected Proteobacteria and Bacteroidota, while Firmicutes, especially *Bacillus*, remained more enriched in the soil inoculum (Supplementary Fig. 4). Exceptions were certain SbRB ASVs (*Pseudomonas*, *Massilia* and *Pantoea*) whose relative abundances were reduced compared with the RhizCom (Fig. 2f). This could be attributed to the inability of certain SbRB to maintain a similar abundance in the RhizCom as a result of the coalescence of the two microbiomes. The changes between the RhizCom and the recovered community (Fig. 2e) are largely attributable to shifts in relative abundances rather than loss of taxa (Fig. 1b and Supplementary Table 3), further emphasizing that the overall RhizCom taxonomic composition remains stable after recovery (Fig. 1g).

### Wheat rhizobiome and source communities functionally differ
The RhizCom, SbRB and initial soil communities were functionally characterized by metagenomic shotgun sequencing. Over 1 billion reads were co-assembled into 1.5 million contigs (>1,000 bp), totalling 5.2 billion bp (Extended Data Fig. 2 and Supplementary Table 4). Sample coverage based on sequence diversity showed nearly complete coverage of the RhizCom and SbRB metagenomes (average coverage >97%), whereas soil coverage was >52%, consistent with higher soil sequence diversity (Extended Data Fig. 2b). The samples showed distinct taxonomic profiles dominated by Bacteria (Supplementary Fig. 5). Among them, Proteobacteria were the most prevalent, representing more than 50% in RhizCom and SbRB samples, but ~20% in the soil. Overall, the taxonomic composition of the metagenomic samples was consistent with that obtained by 16S rRNA amplicon sequencing (Supplementary Fig. 6). The communities differed in their general functional content (ANOSIM, $P = 0.001$, Extended Data Fig. 3) and showed a predominance of genes involved in general cellular functions, including genetic information processing and carbohydrate and amino acid metabolism.

Analysis of the differential abundance of KEGG annotations among the communities and their classification into dominance categories revealed a similar number of enriched functions in both the RhizCom

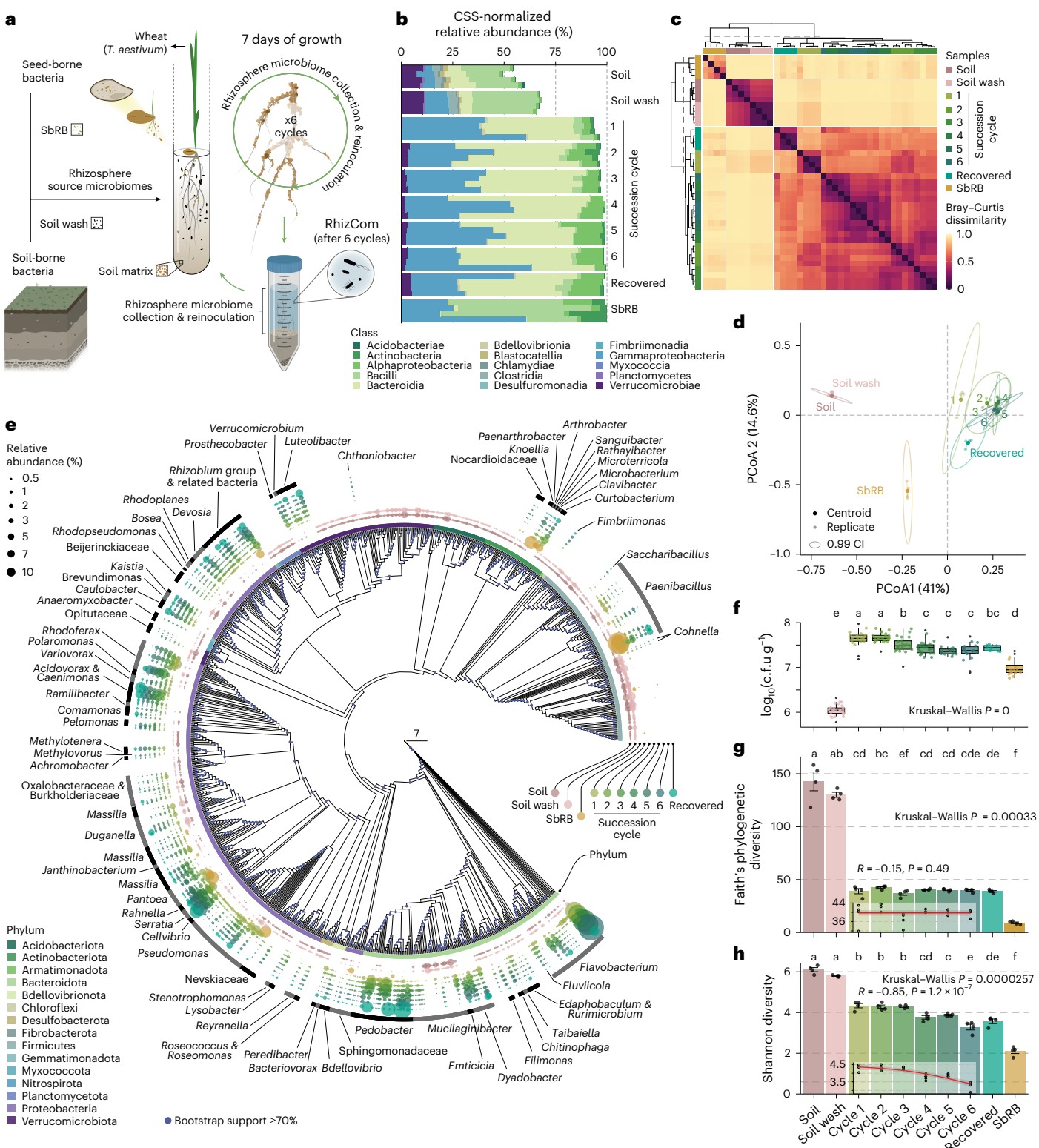

**Fig. 1 | Sequential succession of a wheat rhizosphere microbiome. a**, Experimental design of the microcosm to study the coalescence of soil and seed-borne rhizosphere bacteria (detailed in Extended Data Fig. 1). **b**, CSS-normalized relative abundance of the top 500 ASVs at the class level. **c**, Clustering of Bray–Curtis dissimilarities across sample replicates (*n* = 4 per sample) showed a clear distinction of SbRB, soil, soil wash and succession cycles. **d**, PCoA based on Bray–Curtis dissimilarities across samples. The ellipses represent a 99% confidence interval (CI). **e**, Selection of ASVs during the succession of the wheat rhizobiome. The phylogenetic tree was built using ASVs with a total mean (*n* = 4) relative abundance ≥0.005%. From inwards to outwards, coloured dots represent ASVs, with sizes corresponding to their mean relative abundance in samples. For details see Supplementary Fig. 2. **f**, c.f.u. enumeration of

bacteria on R-2A agar per gram of soil (soil wash *n* = 28) or rhizosphere (cycle 1 *n* = 20; cycle 2 *n* = 21; cycle 3 *n* = 22; cycle 4 *n* = 23; cycle 5 *n* = 15; cycle 6 *n* = 20; recovered *n* = 8; SbRB *n* = 16). The centre line shows the median, the box spans the first to third quartiles, and whiskers extend to 1.5× the interquartile range. Outliers beyond this range are shown as black dots. **g,h**, Measures of Faith's phylogenetic diversity (**g**) and Shannon diversity (**h**) across samples. The mean values are represented by bars (±s.d.). Black dots represent individual replicates (*n* = 4). Different letters indicate significantly different groups (*P* ≤ 0.01) determined using Kruskal–Wallis test, followed by Fisher's post hoc test and FDR *P*-value adjustment. Inset plots show Spearman correlations between alpha diversity measurements and succession cycles (red line, mean values; dots, replicates (*n* = 4)).

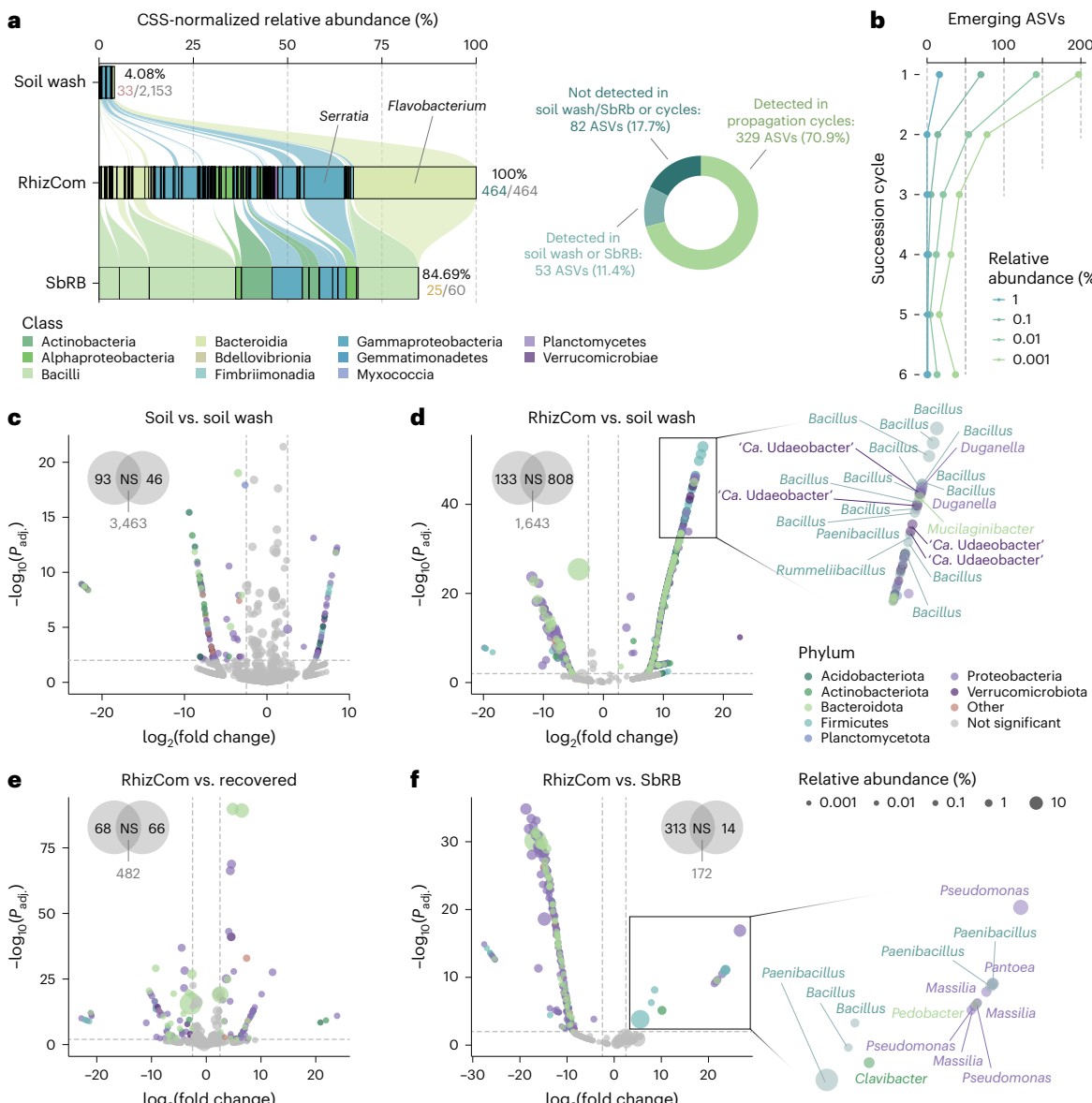

**Fig. 2 | Contribution of SbRB to the RhizCom. a**, RhizCom ASVs traced to the initial soil wash inoculum or the SbRB. Left: alluvial diagram showing the connection of ASVs throughout the 3 samples. Only ASVs present in the RhizCom (6th succession cycle) are considered. Replicates ($n = 4$) were pooled by mean values. Percentages show the combined CSS-normalized relative abundance of the ASVs that contribute to the RhizCom, followed by the number of ASVs/total ASVs in the sample in colour/grey according to the sample. Colour according to ASVs taxonomy at the class level. Right: doughnut chart showing the number of RhizCom ASVs detected in the soil wash or the SbRB. **b**, Emergence of ASVs through the succession cycles. Lines represent the number of new ASVs detected at each cycle at different thresholds of relative abundance. **c**–**f**, Differential ASV abundance calculated with DESeq2 using two-sided Wald test for significance and a local estimate of dispersion. ASVs with a $|\log_2(\text{fold change})| \geq 2.5$ and $P_{\text{adj.}} < 0.01$ were considered as significant. Results are represented as volcano plots of key comparisons. Dots represent ASVs coloured by phylum and sized according to their relative abundance. The number of differentially abundant ASVs per condition is shown within Venn diagrams (NS, not significant). Right of **d** and **f**: zooms with labels on the most depleted ASVs in the RhizCom compared with soil wash or SbRB samples. See Supplementary Fig. 4 for details.

and soil communities (1,989 and 2,012, respectively, Fig. 3a). Enriched functions in the RhizCom were involved in cellular processes, including transport and catabolism, and information processing (Fig. 3b). In contrast, the soil was enriched in functions related to signalling molecules and interaction, and metabolism, including energy metabolism and biodegradation, and metabolism of xenobiotics. SbRB were enriched in membrane transport. Notably, functions with balanced distributions across samples had higher mean abundances than other dominance categories (Fig. 3c and Supplementary Table 5). The enrichment of a category in a given community was not caused by an overall increase in abundance within that community, but rather resulted from lower abundance in the other two communities (Fig. 3c).

**The wheat rhizobiome is enriched in host-interaction traits**

When considering the most enriched functions, iron acquisition was a dominant feature in both the RhizCom and SbRB communities (Fig. 3d). Enriched functions in the RhizCom also included genes for starch and sucrose metabolism, and plant colonization, and beneficial activities such as the biosynthesis of flagella, exopolysaccharides, hydrogen cyanide and 1-aminocyclopropane-1-carboxylate deaminase (Fig. 3d and Supplementary Table 6). The enriched functions in SbRB included genes involved in the transport and metabolism of different saccharides, including cellulose/hemicellulose intermediates. The soil community was enriched in broader substate metabolism and energy production, and the biodegradation of aromatic compounds

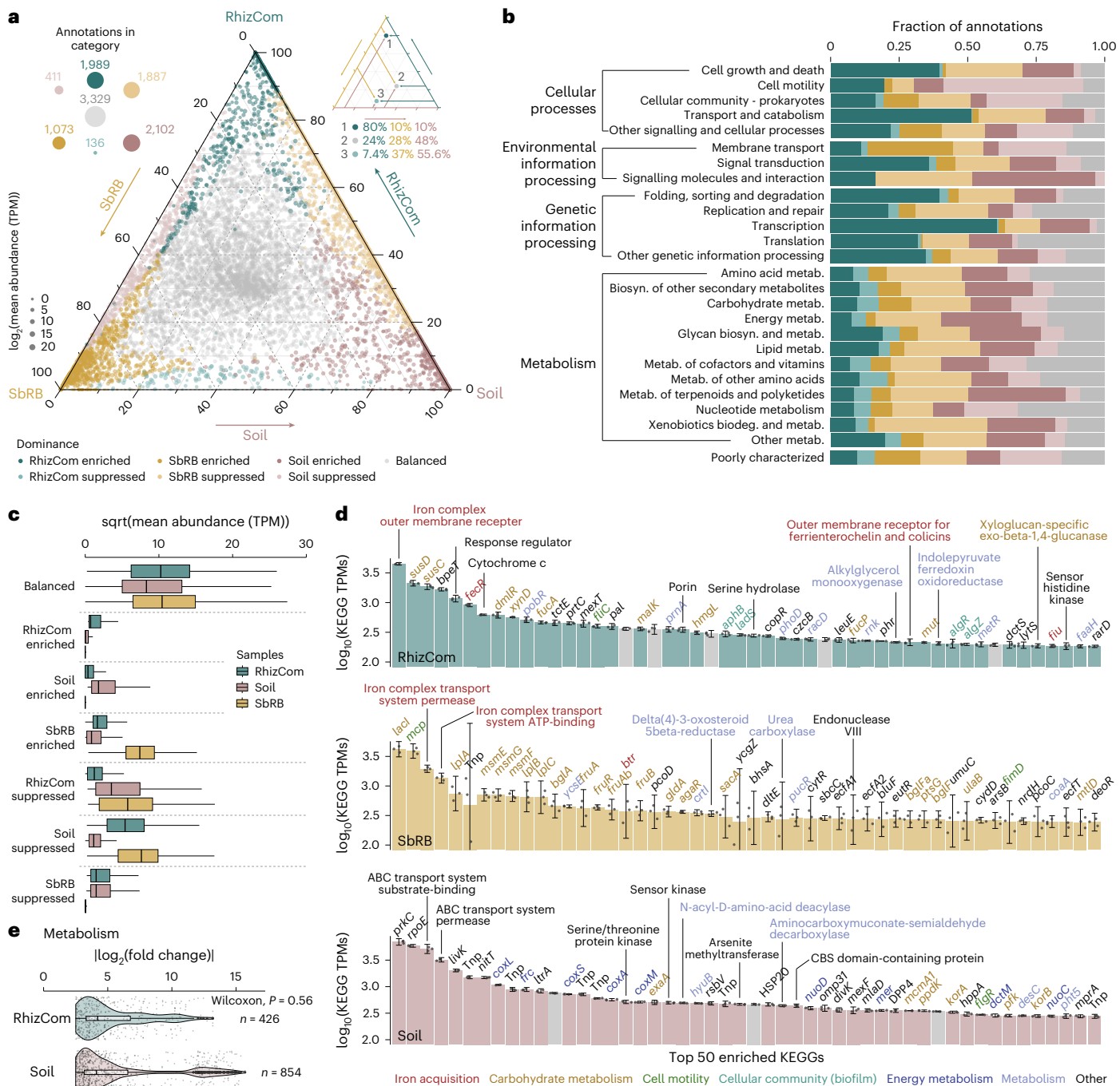

**Fig. 3 | Functional contribution of communities. a**, Ternary plot showing the contribution of each sample to KEGG functional annotation abundances. Each coordinate (represented by dots) consists of the relative contribution of each function in ternary space. Functions are sized according to log₂(mean TPM) across samples. Only annotations with mean TPM > 0.5 were considered. Colour according to dominance categories defined by the result of pairwise differential abundance analyses. Top left: circles and numbers represent the number of functions per dominance category, with sizes at scale. Top right: example of interpretation of function contribution in the ternary space. **b**, Fraction of annotations assigned to the defined dominance categories in **a** per functional KEGG BRITE hierarchy. **c**, Square root (sqrt) of mean KEGG TPMs per sample (*n* = 3 replicates) and dominance category. Statistical differences are shown in Supplementary Table 5. **d**, Abundance (TPMs) of the top 50 functions in samples'

enriched categories. Bars represent mean values (±s.e.). Black dots represent individual replicate values (*n* = 3). Gene names or annotations are indicated. Grey bars indicate KEGGs with unknown functions. Full information across all dominance categories is provided in Supplementary Table 6. **e**, Differentially abundant (|log₂(fold change)| ≥ 2.5 and $P_{adj.}$ < 0.01) KEGG annotations involved in metabolism between RhizCom and soil. Statistical differences were calculated using two-sided Wilcoxon test. The number of differentially abundant annotations is indicated, and they show a higher value in the soil, although the change in abundance (that is, fold change) remains similar (*P* = 0.56). In boxplots (**c**,**d**,**e**), the centre line shows the median, the box spans the first to third quartiles, and whiskers extend to 1.5× the interquartile range. Outliers beyond this range are shown as black dots. Violin plots show the data distribution as a kernel density estimate.

(Fig. 3d and Supplementary Table 6). Furthermore, since SbRB was predominantly integrated into the RhizCom (Fig. 2a), we examined the differential abundance of KEGG annotations between the RhizCom and the soil (Supplementary Table 7). The results emphasized the enrichment of genes involved in biofilm formation, motility and carbohydrate metabolism in the RhizCom, whereas the soil was characterized by genes involved in the biodegradation and metabolism of a wide range of compounds (Fig. 3e and Supplementary Fig. 7). Enriched functions followed a distinct taxonomic distribution within the communities (Extended Data Fig. 4), reflecting the dominant members of each community.

### Specialized SbRB metabolism facilitates rhizobiome assembly

Binning of contigs and dereplication of bins resulted in a total of 821 metagenome-assembled genomes (MAGs), of which 110 were of high to medium quality (completeness >50%, contamination <10%, Fig. 4a). These MAGs represented 22.2% of the RhizCom, 42.8% of the SbRB and 8.4% of the soil communities (Fig. 4b and Supplementary Table 8). The assignment of MAGs to their primary community based on TPMs resulted in 60 MAGs being assigned to the RhizCom, 16 to the SbRB and 33 to the soil communities (Fig. 4c and Supplementary Table 8). The taxonomic classification of the MAGs corresponds to the most abundant taxa on each community. For example, MAGs assigned to SbRB belong to *Pantoea*, *Paenibacillus*, *Rhizobium* or *Pseudomonas* (Figs. 4c and 1e).

Functional differences between the MAGs assigned to the three communities were explored on the basis of KEGG annotations. The results show a higher proportion of annotations for starch and sucrose metabolism (Kruskal–Wallis: $P = 0.00003$) and fructose and mannose metabolism ($P = 0.00019$) in SbRB MAGs (Fig. 4d), as well as a higher number of annotations involved in nicotinate and nicotinamide metabolism ($P = 0.00259$) and riboflavin metabolism ($P = 0.00116$), suggesting a specialized SbRB metabolism. SbRB with the highest number of annotations in these categories belong to *Pantoea* (MAGs M057 and M058), *Paenibacillus* (M008), *Pseudomonas* (M063 and M108) and *Priestia* (M004), consistent with their prevalence (Fig. 1e, Extended Data Fig. 5, and Supplementary Tables 3 and 8).

Specific reactions differentially present in SbRB MAGs revealed their ability to assimilate specific extracellular disaccharides: sucrose, maltose, cellobiose and trehalose via their conversion to the monosaccharides fructose and glucose (Fig. 5). While these upper pathways were absent in members of the soil community, enzymes for lower metabolic pathways (for example, conversion of D-glucose to D-fructose-6P, or D-fructose to D-fructose-6P, Fig. 5b) were similarly encoded by MAGs from the three communities (Fig. 5c). MAGs assigned to SbRB and RhizCom showed a similar prevalence of upper disaccharide assimilation pathways. Most of these reactions were assigned to the most abundant members of SbRB: *Pantoea*, *Paenibacillus* and *Priestia*. In addition, complete pathways for the biosynthesis of nicotinate and riboflavin vitamins ($B_3$ and $B_2$, respectively) were also more prevalent in SbRB MAGs than in soil (6 versus 4 for nicotinate and 8 versus 3 for riboflavin, respectively), with both pathways present in *Pantoea*, *Paenibacillus*, *Priestia* and *Pseudomonas* SbRB MAGs (Fig. 5a). These

results suggest that SbRB initiate the assimilation of disaccharides in the wheat rhizosphere, which can then be utilized by a larger number of RhizCom members.

Saccharide utilization profiles of five isolated RhizCom members were characterized in monocultures and co-cultures. Helper SbRB strains *Pantoea* and *Paenibacillus* enabled partner strains to grow on disaccharides as sole carbon source (Extended Data Fig. 5). This metabolic specialization probably explains their early dominance in the RhizCom, allowing SbRB to efficiently exploit key resources in the developing rhizosphere niche, preemptively consuming simple saccharides for later arrivals.

## Discussion

Crop health and productivity are highly dependent on the complex microbial communities they host[6,7,10]. While recent research highlights the superior role of native natural microbiomes over synthetic communities in supporting key plant functions[36], limitations in obtaining reproducible natural plant-associated microbiomes challenge our ability to experimentally test microbiome manipulation strategies and gain insight into the fundamental principles governing microbial interactions and community dynamics.

In this work, we have demonstrated that sequential propagation of the wheat rhizobiome led to the recruitment of a stable, complex and reproducible microbiome (Fig. 1). This is consistent with previous microbiome steady states achieved after sequential growth in the rhizosphere of different host plants[20], their phyllosphere[37] or soil microbiomes[19], suggesting robust environmentally mediated microbiome selection across hosts and habitats. The pronounced succession effect from the initial soil community to a converging rhizobiome (Fig. 1d,e) results from several host-mediated factors. Carbon sources exuded by young wheat roots[21,22] primarily drove net community growth (Fig. 1f), favouring the proliferation of copiotrophs[38]. We used a 7-day time frame for each microbiome propagation cycle, representing a window of consistently high microbial activity that amplified the selective effect on community assembly. In addition, plant immunity[39,40], the secretion of plant secondary metabolites that influence bacterial assembly[24,25], and the outcome of interbacterial interactions[15,16] probably contributed to the assembly of the rhizobiome, which was exacerbated by successive microbiome propagation cycles. This favoured the assembly of Proteobacteria and Bacteroidetes on the wheat roots to the detriment of soil-dwelling Acidobacteria or Planctomycetes (Fig. 1e), as previously observed[41,42]. The resulting RhizCom was resilient to cryopreservation and regrowth on wheat roots, effectively maintaining a stable and reproducible rhizosphere community.

SbRB had a leading representation in the RhizCom after coalescing with members of the soil microbiome, most of which were acquired during cycles of community succession (Fig. 2 and Supplementary Table 3). Although the SbRB community was based on a limited set of plants during a single propagation cycle, it was sufficient to capture 50% of the relative abundance of the RhizCom. Moreover, integrating the information from the succession cycles, which accounted for a total of 96 individual plants, 91% of the relative abundance of

**Fig. 4 | Characteristics of metagenome-assembled genomes. a**, MAG quality based on percentage of completeness and contamination. High-quality MAGs: >90% completeness, <5% contamination; good-quality MAGs: >75% completeness, <10% contamination; medium-quality MAGs: >50% completeness, <10% contamination; low-quality MAGs: ≤50% completeness, <10% contamination; discarded MAGs: ≥10% contamination. Dashed grey lines indicate the MAG quality thresholds. The number and percentage of MAGs per quality category are shown in the doughnut plot. **b**, Percentage of the total number of reads mapped to MAGs. Unmapped and unclassified reads were not considered for calculation of percentages. **c**, Taxonomic classification of high- to medium-quality MAGs and characteristics. Taxonomic classification was based on the last taxonomic rank as obtained by MAG contig consensus. Phylogenetic tree based on maximum-likelihood (ML) of concatenated conserved amino acid markers, expanding 38,640 aligned positions and using 100 rapid bootstrap inferences followed by a thorough ML search for the best-scoring tree. Seven MAGs lacking sufficient marker gene sequences for phylogenetic inference were discarded. Strain het., strain heterogeneity. **d,e**, Metabolic features enriched in SbRB MAGs. Number of annotations in fructose and mannose and starch and sucrose (**d**) or nicotinate/nicotinamide (**e**) metabolic categories. The centre line shows the median, the box spans the first to third quartiles, and whiskers extend to 1.5× the interquartile range. Outliers beyond this range are shown as individual points.

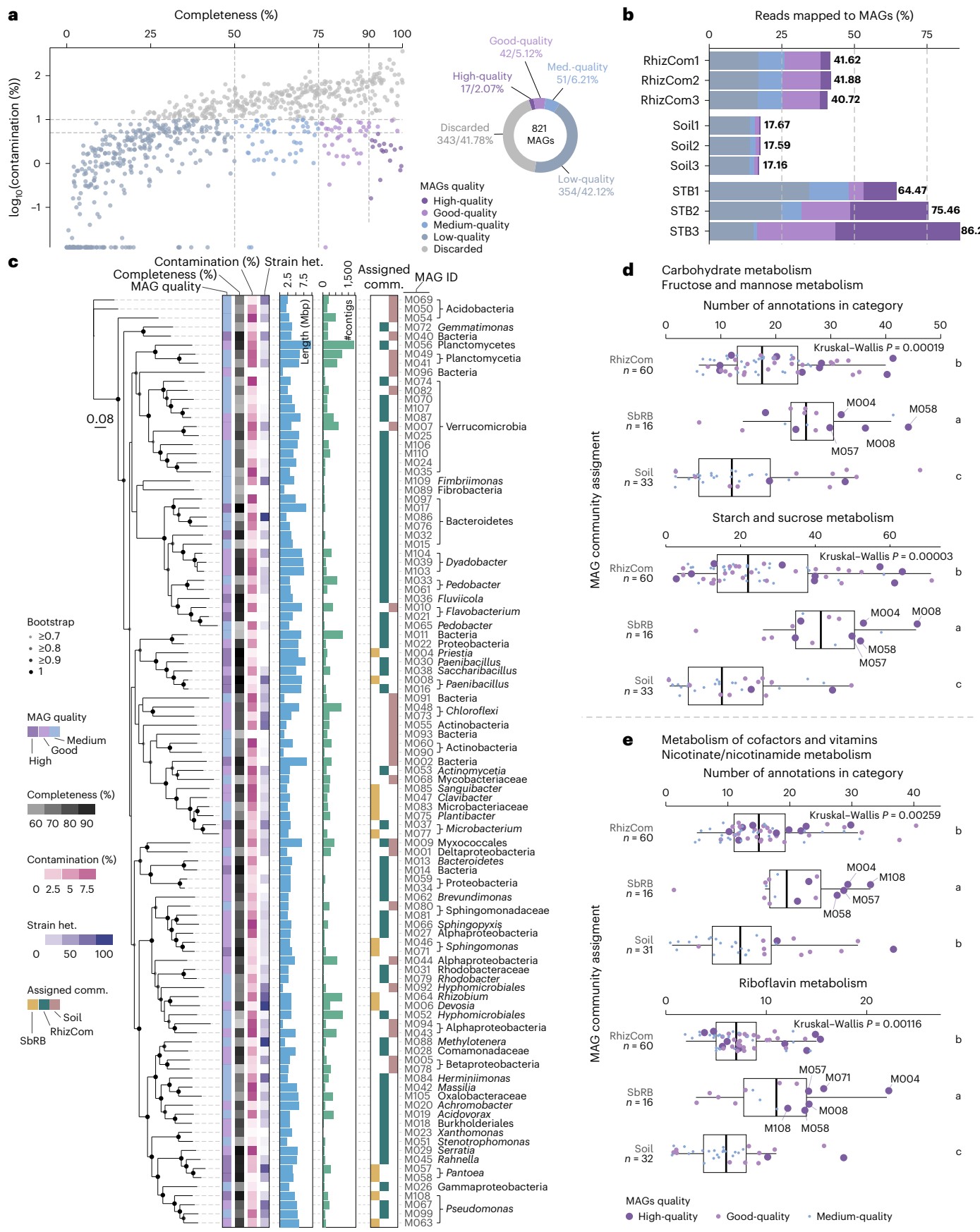

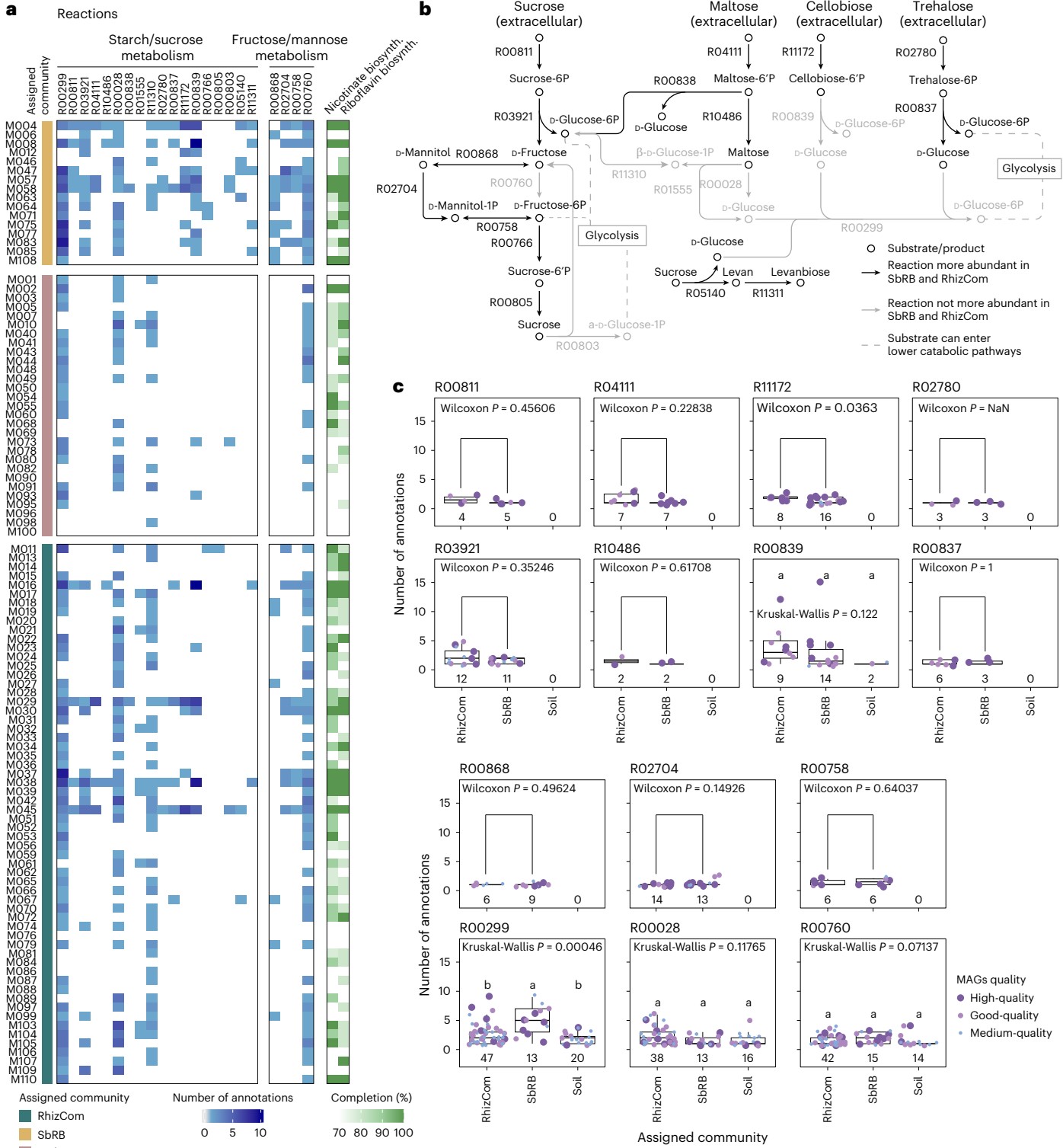

**Fig. 5 | Metabolism of saccharides by SbRB drives the RhizCom assembly.**
**a**, Distribution of major KEGG reactions involved in specific saccharide metabolism within communities' MAGs (blue) or the biosynthesis of nicotinate and riboflavin vitamins (green). Nicotinate: R07407, R00481 + R04292 + R03348 + R03346 + R02295. Riboflavin: R00425 + R03459 + R03458 + R07280 + R07281 + R04457 + R00066 + R00549 + R00161. **b**, Reactions in pathways. Circles represent substrates and products metabolized by enzymes (arrows). Dashed lines represent substrates that can enter lower catabolic pathways (that is, glycolysis). **c**, Number of annotations belonging to key

KEGG reactions per MAG and assigned community. Numbers below the boxplots indicate the number of MAGs that contain the reaction. Upper catabolic pathways of specific disaccharides (sucrose, maltose, cellobiose and trehalose) are specific to SbRB/RhizCom members and not present in soil MAGs, while lower catabolic pathways (R00299, R00028, R00760) are present in most MAGs. NaN, *P* value not computed due to insufficient variability between groups. The centre line shows the median, the box spans the first to third quartiles, and whiskers extend to 1.5× the interquartile range. Outliers beyond this range are shown as individual dots.

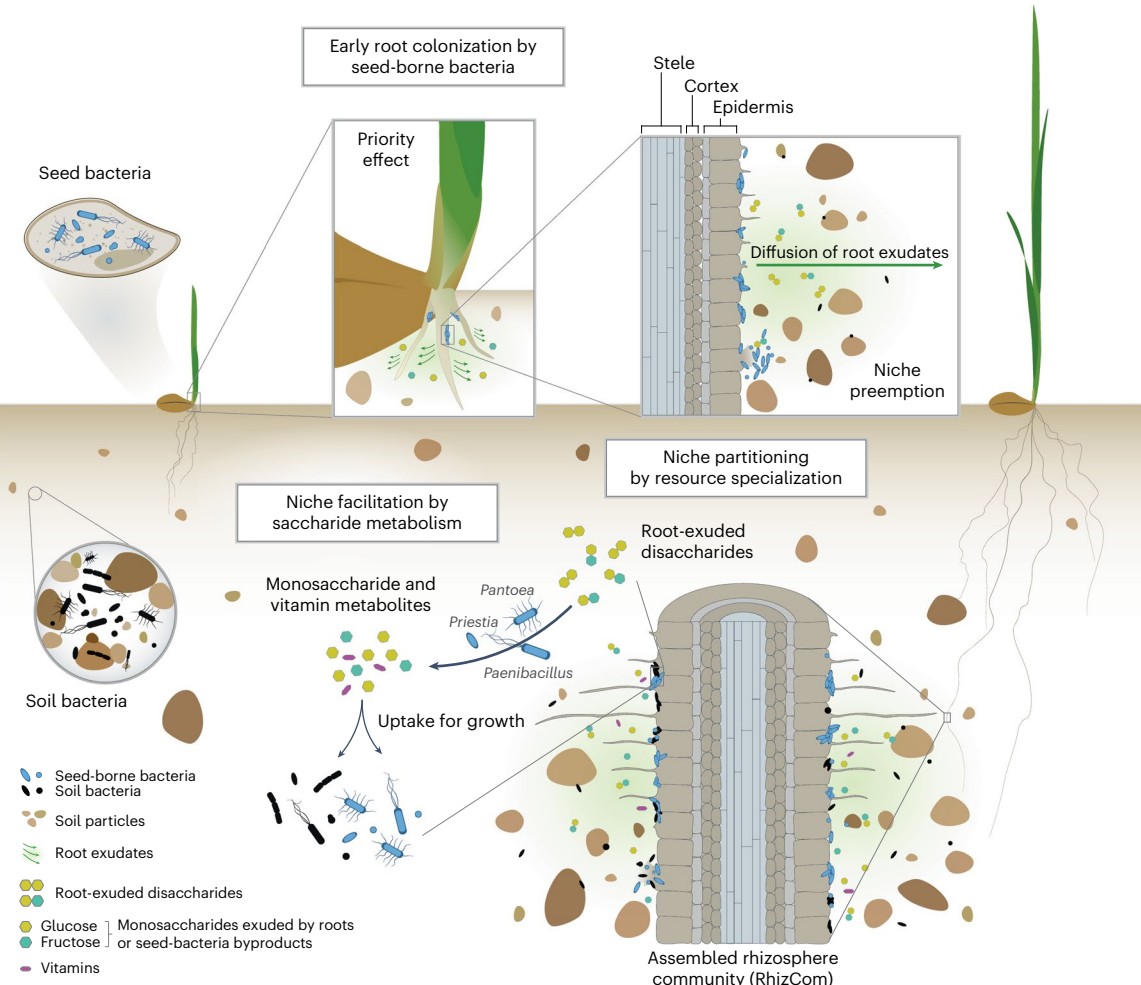

**Fig. 6 | Priority effects of heritable seed-borne bacteria drive early assembly of the wheat rhizosphere microbiome.** Representation of the processes that occur during early assembly of the wheat rhizosphere microbiome. Upon germination, seed bacteria are the first to colonize root surfaces, using root exudates as carbon sources. This early colonization leads to the depletion of readily available nutrients in the rhizosphere (niche preemption), preventing early colonization by soil-dwelling bacteria. In later stages of plant development, the specialized metabolism of root-exuded disaccharides by seed-borne bacteria allows them to persist when competition for primary resources from late-arriving soil bacteria increases through niche partitioning. The resulting accumulation of disaccharide metabolism by-products, glucose and fructose, by SbRB facilitates the colonization of the wheat rhizosphere by both soil and seed-borne bacteria. In addition, the biosynthesis of the vitamins nicotinate and riboflavin by SbRB supports the establishment of auxotrophic bacteria, promoting community coalescence and increasing the complexity of the rhizosphere microbiome.

RhizCom was seed borne. The dominance of seed-transmitted bacteria in the rhizobiome of various young plants has been observed previously[28], with their influence varying across different soil microbiome compositions[43], and probably reflects instances of priority effect where pioneer members of the seed microbiota strongly shape the root-associated community[44,45]. We argue that the rapid habitat changes during early stages of plant development and the periodic resetting to the initial habitat state further stressed primary succession. This effect would have strongly favoured the assembly of bacteria adapted to the rapid utilization of carbon exuded by developing roots and may have limited the establishment of late-arriving soil bacteria, for example, by niche preemption[44], although this could still occur at later stages of habitat development (Fig. 6). Nevertheless, 8% of the relative abundance of the RhizCom was derived from the soil community. This could be explained by some members having similar metabolic capabilities to SbRB, strong competitive abilities, or could be evidence for instances of niche facilitation[46] (for example, metabolic by-products of one bacterium facilitating the growth and establishment of another). Whether these processes will maintain the reproducibility of the RhizCom in later stages of habitat development[47] remains to be investigated. Our microcosms may introduce biases compared with natural conditions, such as the lack of soil heterogeneity and the use of surface-disinfected seeds, which in nature carry bacteria that may contribute to rhizobiome assembly.

The patterns of functional specialization observed among the communities (Fig. 3) were related to the exploitation of their different habitats. In the soil community, enriched functions characteristic of oligotrophic bacteria[41,42] such as Actinobacteria or Planctomycetes, reflected their broad energy metabolism and secondary metabolite production (Fig. 3b, Extended Data Fig. 4 and Supplementary Fig. 7), supporting survival in resource-limited, variable environments[42,48,49]. This broad metabolic capacity was further highlighted by the higher metabolic diversity of the soil community compared with the RhizCom (Fig. 3e). In contrast, the RhizCom exhibited a functional profile dominated by enrichment in transcription, signal transduction and other traits characteristic of metabolically active copiotrophs[38,50]. In addition, traits required for successful colonization and resource competition in the rhizosphere were enriched in the RhizCom, including biofilm formation, motility, or carbohydrate and amino acid metabolism[51–53] (Supplementary Fig. 7). The biosynthesis of extracellular polymeric levan may have additionally supported the effective colonization of the rhizosphere[54] by SbRB (Fig. 5).

In contrast to the other two communities, SbRB did not show specific enrichment in any of the broad functional categories, except for membrane transport (Fig. 3a,b). This probably reflects that the functional traits of SbRB were largely integrated into RhizCom during sequential community succession. However, specific enriched functions in SbRB were related to carbohydrate metabolism, such as sugar transport and metabolism systems (*msmEFG*, *bglAF*, *fruAB*, Fig. 3d) and, together with the RhizCom, also iron acquisition. The enrichment of these functions in SbRB members corresponds to essential traits in early instances of habitat exploitation by the seed microbiota, including the utilization of saccharides exuded by developing plant roots[53,55] and scavenging strategies to secure limiting iron[56,57]. Indeed, the resource specialization of SbRB in the metabolism of sucrose, maltose, cellobiose and trehalose (Fig. 5)—disaccharides commonly exuded by plant roots or produced during cellulose degradation[53]—emphasizes their role as early metabolic specialists, particularly *Pantoea* and *Paenibacillus*. Since disaccharides are less readily available carbon sources compared with glucose and fructose, which are also exuded by wheat roots[58,59], SbRB may have initially focused on the exclusive use of monosaccharides in the absence of early soil-dwelling competitors due to priority effects, leading to the preemption of simple sugars[44]. As competition for primary resources increases with later-arriving bacteria, specialized disaccharide metabolism by established SbRB would have allowed them to persist in the rhizosphere through niche partitioning. The monosaccharide by-products of this metabolism could then accumulate, in turn facilitating broader bacterial colonization, suggesting a secondary role for niche facilitation (Fig. 6). This sequence of niche partitioning and facilitation by saccharide metabolism allows SbRB to maintain their dominance in the rhizosphere at later stages of plant development. By producing the vitamins nicotinate and riboflavin (Fig. 5a), SbRB may have further contributed to niche construction, enabling both soil bacteria and other seed-borne members lacking these functions to coalesce into the RhizCom[60] (Fig. 6). This study highlights the key role of SbRB in shaping early rhizobiome assembly, as demonstrated by their ability to facilitate the growth of other rhizosphere bacteria through disaccharide metabolism (Extended Data Fig. 5). Future work should investigate whether these processes extend to other plant species and cultivars.

Our work demonstrates that sequential propagation is a key strategy for achieving reproducible rhizobiomes, driven by strong host-mediated effects under constrained habitat conditions. Priority effects and niche facilitation through saccharide metabolism of seed-borne bacteria were key determinants for the assembly of the rhizosphere community, highlighting the processes that drive primary succession during early habitat development. These results contribute to our understanding of microbial community dynamics and provide valuable strategies for experimentally testing microbiome manipulation approaches aimed at improving crop productivity and health.

## Methods

### Initial soil sampling and characterization

The soil used for the wheat rhizobiome recruitment was collected from a mixed plant species grassland bordering a wheat field in the region of Grandcour, Switzerland (46.884947° N, 6.922562° E) in April 2021, as previously described[10]. The bacterial load of the soil was determined following mixing of soil and mineral medium[18] (MM, 1:1). The mixture was incubated for 1 h at room temperature with shaking (180 r.p.m.) and then centrifuged at $300 \times g$ for 1 min to pellet soil debris. The supernatant was further centrifuged at $4,500 \times g$ for 5 min. The resulting pellet was washed twice with MM. Serial dilutions were plated onto R-2A medium (Millipore) and incubated at 25 °C. Colony-forming units (c.f.u.s) were enumerated after 48 h. Three independent samples and 10 technical replicates per sample were performed, resulting in an average of $1.18 \times 10^6 \pm 3.43 \times 10^5$ c.f.u.s g$^{-1}$ of soil. Raw data were recorded in Microsoft Excel (v.2409 Build 16.01.18025.20160).

### Sequential propagation of the wheat rhizobiome

To obtain a reproducible wheat rhizobiome, we used a previously designed microcosm[18]. Each microcosm consisted of a sterile 3.3 borosilicate glass tube (diameter 25 mm × 200 mm, Fisher Scientific) closed with sterile Magenta 2-way polypropylene caps (Sigma-Aldrich). Each microcosm contained a soil matrix, soil extract, a bacterial inoculum and a pregerminated 2-day-old wheat *Triticum aestivum* cv. Arina seedling (Extended Data Fig. 1).

The soil matrix, which served as a substrate for plant root and microbiome development, comprised 20 g of 0.5–4 mm silt that had been sieved and autoclaved twice (120 °C, 15 min). The soil extract, which provided micronutrients and minerals from the soil environment, was prepared by mixing soil with MM medium (1:1 ratio) for 1 h at 180 r.p.m., followed by centrifugation at $4,500 \times g$ for 15 min. The resulting supernatant was filtered through 0.22 µm cellulose acetate filters (Dutscher) and autoclaved twice, and its final pH was 6.48 ± 0.03. The bacterial inoculum was either an initial soil wash containing a bacterial suspension extracted from the soil, or a rhizosphere wash obtained after each re-inoculation cycle (see below). The soil wash was prepared by mixing soil with MM (1:1 ratio) for 1 h at 180 r.p.m., followed by a low-speed centrifugation at $300 \times g$ for 1 min to remove soil debris. The supernatant was then centrifuged at $4,500 \times g$ for 15 min, and the resulting pellet washed twice with MM. The pellet was resuspended in soil extract and considered as the soil wash suspension containing cells from soil. Of the soil wash, 2 ml was added to the 20 g of soil matrix in each glass tube. Finally, a surface-disinfected pregerminated wheat seedling was placed in each tube. For pregermination, seeds were surface disinfected in 4% NaClO for 15 min, thoroughly washed with sterile distilled water and germinated on 0.85% (w/v) agar plates in the dark at 22 °C for 2 days. The microcosms were incubated in a Percival PGC-7L2 plant growth chamber at 22/18 °C, 16/8 h light/dark photoperiod (light intensity, 160 µE m$^{-2}$ s$^{-1}$) and 70% relative humidity. After 7 days of plant growth in the microcosm, the plants were removed from the tubes and their roots were gently shaken to detach any loosely adhering particles of soil matrix. Roots with the adhering soil matrix were cut off at the point of emergence from the seed, representing the rhizosphere, which also encompasses the rhizoplane. The rhizospheres of four plants were pooled and weighed. A volume of 25 ml of MM was added to each sample pool, followed by vortexing for 20 min. The samples were then centrifuged at $300 \times g$ for 1 min to pellet soil matrix debris, and the resulting supernatants were considered the rhizosphere wash, which was used for total DNA extraction, c.f.u. enumeration, and to prepare the re-inoculation solution. The re-inoculation solution was prepared by mixing 10 ml of the rhizosphere wash of 4 replicates (total of 40 ml of rhizosphere wash), which was then centrifuged at $4,500 \times g$ for 5 min. The supernatant was discarded and the pellet was resuspended in 40 ml of soil extract. This solution was used to inoculate the next cycle. The microcosms were maintained in a Percival PGC-7L2 plant growth chamber at 22/18 °C, 16/8 h light/dark photoperiod (160 µE m$^{-2}$ s$^{-1}$ light intensity) and a relative humidity of 70% for 7 days. Sixteen tubes were prepared per propagation cycle. After 7 days, the rhizosphere wash of pools of 4 plants was recovered and used for c.f.u. enumeration of bacteria, total DNA extraction and for the re-inoculation of the next cycle. For c.f.u. enumeration of bacteria, aliquots of the rhizosphere wash were serially diluted in MM and plated on R-2A plates. Eight technical replicates per pool of 4 plants were used for c.f.u. enumeration. For DNA extraction, 10 ml of the rhizosphere wash was centrifuged at $7,000 \times g$ for 15 min to pellet bacterial cells, which were cryopreserved at −20 °C until DNA extraction (see below). Finally, another 10 ml of the 4 rhizosphere washes of all 4 rhizosphere pools were combined into a single suspension, which was centrifuged at $7,000 \times g$ and the supernatant discarded. The pellet was resuspended in 40 ml of soil extract, and 2 ml of the suspension was inoculated per tube for the next propagation cycle. A total of six propagation cycles were performed, after which the

final rhizosphere wash was used for c.f.u. enumeration of bacteria and total DNA extraction as described above, and to preserve the microbial community by storing the suspension at −80 °C in 50% (v/v) glycerol.

The cryopreserved wheat rhizosphere community stored at −80 °C was recovered by thawing and centrifuging two aliquots at 7,500 × *g* for 1 min to pellet the cells, and then washing the pellet twice with MM. Finally, the pellet was resuspended in 40 ml of soil extract which served to inoculate 16 microcosms with wheat seedlings, as described. To recover the seed-borne rhizosphere bacteria (SbRB) community, 16 microcosms were set up for a cycle (7 days) of plant growth without addition of soil inoculum. After 7 days, the rhizospheres were collected, pooled by 4 and processed as described above.

### DNA extraction, library preparation and sequencing

Total DNA from samples was extracted using the DNeasy Power-Soil Pro kit (Qiagen), following manufacturer specifications. DNA concentrations were measured using the Qubit dsDNA HS assay kit (Invitrogen). Libraries of the 16S rRNA gene were prepared using 10 ng of extracted DNA per PCR reaction, using primers targeting the 16S rRNA V3–V4 region following the Illumina 16S Metagenomic Sequencing Library protocol, as previously described[18]. The A Nextera XT index kit (v.2, Illumina) was used for indexing. Samples were then quantified and pooled in equal amounts for sequencing. Samples were spiked with 25% PhiX control DNA and paired-end sequenced using an Illumina MiSeq v.3 instrument running for 600 cycles. For shotgun metagenome sequencing, samples from the initial soil, after the sixth re-inoculation cycle, and SbRB were selected. A total of 100 ng of DNA per sample were used for library preparation, using the Nextera DNA Flex library protocol (Illumina). Metagenome samples were paired-end sequenced using a NovaSeq 6000 instrument running for 300 cycles and using S2 flow cells. Samples were sequenced at the Lausanne Genomic Technologies Facility (Lausanne, Switzerland). The raw reads from both 16S rRNA gene amplicons and shotgun metagenome were quality filtered and trimmed with fastp[61] v.0.23.2 using the adaptor autodetection option.

### 16S rRNA gene amplicon analyses

Quality-filtered 16S rRNA gene amplicon reads were analysed following the DADA2 (ref. [62]) v.1.30.0 pipeline, as previously described[18]. SILVA database[63] v.138.1 was used for taxonomy assignment at 99% sequence identity. ASV sequences were used to construct a maximum-likelihood (ML) phylogeny as previously described[18]. Data was imported into the phyloseq[64] v.1.46.0 R package. ASVs classified as mitochondria, chloroplasts, or with prevalence <0.05 across samples were removed. Cumulative sum scaling (CSS) normalization[65] was applied to ASV counts. Relative abundances at the bacterial class taxonomic rank were analysed and represented within the 500 most abundant ASVs using the 'plot_bar' function within the phyloseq R package. Observed ASVs and the Shannon diversity index were calculated and plotted with the 'plot_richness' phyloseq R function. Phylogenetic diversity (Faith's PD) was calculated using the 'pd' function within the picante v.1.8.2 R package[66]. A principal coordinate analysis (PCoA) was performed with the 'ordinate' phyloseq R function and using Bray–Curtis dissimilarities. Bray–Curtis dissimilarities were also used for a complete-linkage hierarchical clustering using the ComplexHeatmap[67] v.2.20.0 R package. Differential abundance analysis of ASVs across samples was performed using the DESeq2 (ref. [68]) v.1.44.0 R package as previously described[18]. The phylogenies of ASVs were visualized using the ggtree[69] v.3.12.0 and ggtreeExtra[70] v.1.14.0 R packages.

### Metagenome analyses

Quality-filtered reads were mapped against a reference wheat genome (NCBI GenBank acc. no. GCA_903993985.1) to remove host contamination using Bowtie2 (ref. [71]) v.2.5.1. SAMtools[72] v.1.7 was used to retrieve unmapped reads. To assess sample coverage, Nonpareil3 (ref. [73])

v.3.401 was used with the host-removed reads, using a *k*-mer of 31 and 100,000 random reads. Reads were downsampled to a maximum of 0.6 billion reads per replicate to account for computational limitations, and subsequently normalized by *k*-mer coverage using BBnorm v.37.62 (sourceforge.net/projects/bbmap). A first normalization run was performed using a target *k*-mer coverage of 50 per replicate paired-end read. Reads from the sample replicates were then pooled, and a second normalization run also using a target *k*-mer coverage of 50 was performed. To reduce the complexity of the soil sample, pooled reads with a *k*-mer coverage below 15 were discarded. The number of reads obtained at each step was assessed with Seqkit[74] v.2.2.0 as summarized in Extended Data Fig. 2. Normalized pooled reads from the last re-inoculation cycle (RhizCom), SbRB and soil were co-assembled using MEGAHIT[75] v.1.2.9 with the options '–presets meta-large and–kmin-1pass' and a minimum contig length of 1,000 bp. The assemblies were analysed using QUAST[76] v.5.2.0, resulting in 1.54 million contigs totalling 5.2 billion bp (Extended Data Fig. 2).

The resulting contigs were analysed using the SqueezeMeta[77] v.1.6.2 pipeline. Briefly, open reading frames (ORFs) were predicted using Prodigal[78] v.2.6.3, followed by similarity searches using the GenBank non-redundant (nr)[79], eggNOG[80] and KEGG[81] databases with DIAMOND[82] v.2.0.15.153. Gene searches were also performed using HMMER3 (ref. [83]) v.3.4 against the Pfam[84] database. The databases were downloaded in May 2023 using the script 'make_databases.pl' from SqueezeMeta. The least common ancestor (LCA) algorithm[77,85] was applied to assign genes at different taxonomic ranks using the search results against the GenBank nr database. Reads after host removal were used for mapping against contigs using Bowtie2 (ref. [71]) v.2.5.1 and to calculate coverage and abundance estimates (normalized transcripts per kilobase million, TPM) for genes. Results were analysed using the SQMtools[86] v.1.6.2 R package.

For differential abundance analyses, we defined a minimum abundance threshold ≥1,000 across samples to be included in the analysis (Supplementary Fig. 8), using DESeq2 as specified above. KEGG annotations were represented in a ternary plot using the ggtern[87] v.3.5.0 package. Mean TPMs per sample were used to calculate ternary coordinates per KEGG annotation, with a minimum TPM threshold ≥0.5 across samples. Differential abundance results were used to categorize a KEGG annotation as enriched if the KEGG annotation was significantly more abundant in a sample in all comparisons that included that sample, or as suppressed if the annotation was not significantly more abundant in any of the comparisons. Intermediate cases and those where the annotation was not significantly enriched in two or three comparisons were considered as balanced.

### Analyses of MAGs

Contigs were binned into MAGs using CONCOCT[88] v.1.1.0, MetaBAT 2 (ref. [89]) v.1:2.15 and MaxBin[90] v.2.2.6. Resulting bins were dereplicated and aggregated into a single set of MAGs using DASTool[91] v.1.1.1. The coverage of MAGs across samples was estimated by mapping host-removed reads to bins, as described above. MAGs statistics were calculated using CheckM[92] v.1.1.6, and taxonomic assignment was based on the bin gene consensus from SqueezeMeta[77]. The quality of MAGs was assessed according to standard metrics[93]. The ML phylogeny of 110 high- to medium-quality MAGs was built with PhyloPhlAn[94] v.3.0.67 using its database of 400 universal amino acid marker sequences[95], and the '–diversity medium' option. Bootstrap support was assessed using RAxML[96] v.8.2.12 with 1,000 rapid bootstrap inferences, followed by a thorough ML search for the best-scoring tree. Results were imported into R and plotted using the ggtree and ggtreeExtra R packages. Seven MAGs without sufficient PhyloPhlAn marker sequences for inclusion in the phylogeny were discarded. A MAG was assigned to a community if its mean TPM percentage was ≥1.5× that of the other two communities. If this condition was not met, the MAG was assigned to communities where the mean TPM percentage difference was <1.5×.

## Bacterial isolation and saccharide utilization profiling

Bacterial strains *Variovorax* sp. DGS2, *Acidovorax* sp. DGS4, *Pseudomonas* sp. DGS16 and *Paenibacillus* sp. DGS31 were isolated from the RhizCom as described previously[97]. Strain *Pantoea* sp. SbRB3 was isolated from the seed-borne rhizosphere bacteria (that is, after 7 days of plant growth in microcosm conditions without addition of bacterial inoculum). Taxonomic classification was based on the partial 16S rRNA sequence using the 27F and 1942R primers[98]. The strains were selected from a larger RhizCom collection of isolates on the basis of growth rate and phenotypic differences that allowed their differentiation when co-cultured and grown on R-2A agar.

For saccharide utilization profiling, strains were grown overnight in M9 base media (per litre: 6 g of $Na_2HPO_4$, 1.5 g of $KH_2PO_4$, 0.25 g of NaCl and 0.25 g of $NH_4Cl$) supplemented with 30 mM of D-glucose monohydrate. The medium was supplemented with the following micronutrients (per litre): 0.29 g of $MgSO_4 \cdot 7H_2O$, 0.067 g of $CaCl_2 \cdot 2H_2O$, 0.00002 g of $(NH_4)_6MoO_{24} \cdot 4H_2O$, 0.002 g of $FeSO_4 \cdot 7H_2O$, 0.2 g of nitriloacetic acid trisodium salt monohydrate, 0.192 g of $Na_2EDTA \cdot 2H_2O$, 0.548 g of $ZnSO_4 \cdot 7H_2O$, 0.457 g of $FeSO_4 \cdot 7H_2O$, 0.077 g of $MnSO_4 \cdot H_2O$, 0.0196 g of $CuSO_4 \cdot 5H_2O$, 0.0124 g of $Co(NO_3)_2 \cdot 6H_2O$ and 0.0089 g of $Na_2B_4O_7 \cdot 10H_2O$. In addition, the following vitamins were added to the medium (per litre): 400 μg of thiamine HCl, 500 μg of D-pantothenic acid Ca salt, 400 μg of riboflavin, 150 μg of nicotinic acid, 100 μg of biotin, 40 μg of folic acid, 200 μg of *p*-aminobenzoic acid, 200 μg of pyridoxamine HCl and 100 μg of lipoic acid. The cells were then centrifuged at $5,500 \times g$ for 5 min and washed twice with MM. The optical density at 600 nm ($OD_{600}$) was adjusted to 0.001 and 5 μl of bacterial suspensions were inoculated into 96-well plates containing 195 μl of M9 medium supplemented with each of the tested saccharides normalized to 30 mM of carbon. The saccharides tested were: D-glucose monohydrate, D-fructose, sucrose, trehalose, maltose and cellobiose. Plates were incubated at 25 °C for 24 h. For co-cultures, 5 μl of each strain at $OD_{600} = 0.001$ were inoculated into 190 μl of M9 medium supplemented with each saccharide. After 24 h of incubation, cells were serially diluted in MM and plated on R-2A agar. Monocultures were plated by dropping 5 μl of diluted cell suspensions onto R-2A agar, while co-cultures used 10-μl drops which were allowed to slide on the plate before drying in a laminar hood. Plates were incubated at 25 °C for 48 h, with c.f.u. appearance monitored every 12 h and counted when clearly distinguishable. Experiments were performed in triplicate. Controls containing M9 medium without any carbon source were used for growth normalization, that is, their mean values were subtracted from the bacterial growth in the presence of saccharides. Raw c.f.u. data are shown in Supplementary Table 9.

## Statistics and reproducibility

Data used for statistical analyses were tested using the Shapiro–Wilk normality test and the Levene's test for homogeneity of variances. Data that did not have a normal distribution or homogeneous variances were analysed with the agricolae[99] v.1.4-5 R package using the non-parametric Kruskal–Wallis rank-sum test, with Fisher's least significant difference (LSD) post hoc criterion and correction of *P* values using false discovery rate (FDR). Pairwise comparisons were performed using two-sided Wilcoxon rank-sum tests within the stats v.4.4.1 R package. For testing increases in bacterial growth when co-cultured with helper strains, one-sided Wilcoxon rank-sum tests were used, with the 'greater' alternative hypothesis. Data that followed a normal distribution and with homogeneous variances were analysed using permutational multivariate analyses of variance (PERMANOVA) or analysis of similarity (ANOSIM), using with the adonis2 or anosim functions, respectively, within the vegan[100] v.1.5-4 R package and using 9,999 permutations. Spearman correlations were calculated using the 'stat_cor' function within the ggpubr R package, and data were fitted to a general additive model using $k = 3$ with the 'geom_smooth' ggplot2 v.3.5.1 R function and a confidence interval of 0.95. No statistical method was used to predetermine sample size. No data were excluded from the analyses. The experiments were not randomized. The investigators were not blinded to allocation during experiments and outcome assessment.

## Reporting summary

Further information on research design is available in the Nature Portfolio Reporting Summary linked to this article.

## Data availability

Raw reads from both 16S rRNA gene amplicons and metagenomic shotgun have been deposited in the NCBI Sequence Read Archive (RSA) database and are publicly available under BioProject accession number PRJNA1169405. The 16S rRNA gene sequences of the five isolated bacterial strains reported in this work have been submitted to the NCBI GenBank database under the following accession numbers: *Variovorax* sp. DGS2, PQ776219; *Acidovorax* sp. DGS4, PQ776220; *Pseudomonas* sp. DGS16, PQ776221; *Paenibacillus* sp. DGS31, PQ776222; and *Pantoea* sp. SbRB3, PQ776223. Other raw data generated in this study are provided in Supplementary Information or in the GitHub repository (https://github.com/dgarrs/RhizCom), and are available on Zenodo at https://doi.org/10.5281/zenodo.13969370 (ref. 101).

Databases and datasets used in this study are available as follows: wheat reference genome (*Triticum aestivum*, NCBI GenBank acc. no. GCA_903993985.1); SILVA database[63] v.138.1 (https://www.arb-silva.de/documentation/release-138.1/); databases GeneBank nr[79], eggNOG[80], KEGG[81] and Pfam were downloaded using the script make_databases.pl from SqueezeMeta; PhyloPhlAn database of 400 universal amino acid marker sequences[95] was downloaded using the phylophlan_setup_database script of PhyloPhlAn. All databases were downloaded in May 2023. Source data are provided with this paper.

## Code availability

The code used for the analysis of the 16S rRNA gene amplicon data, metagenomic shotgun data, and other analyses reported in this study is available on GitHub (https://github.com/dgarrs/RhizCom) and on Zenodo at https://doi.org/10.5281/zenodo.13969370 (ref. 101).

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

## Acknowledgements

We thank J. Vorholt and A. Pacheco at ETH Zurich for useful discussions and feedback on the paper; J. Vacheron for valuable input during research discussions; C. Matasci (Delley Seeds and Plants Ltd, Switzerland) for providing us with the wheat seeds used in this study; the Lausanne Genomic Technologies Facility in Lausanne, Switzerland for the sequencing service. This work was supported as part of the NCCR Microbiomes, a National Centre of Competence in Research, funded by the Swiss National Science Foundation (grant numbers 180575 and 225148 to C.K.). D.G.-S. received funding from the Fondation pour l'Université de Lausanne.

## Author contributions

D.G.-S. and C.K. conceived and designed the experiments. D.G.-S. performed the experiments, analysed the data and contributed materials/analysis tools. D.G.-S. and C.K. wrote the paper.

## Funding

## Competing interests

The authors declare no competing interests.

## Additional information

**Extended data** is available for this paper at https://doi.org/10.1038/s41564-025-01973-1.

**Correspondence and requests for materials** should be addressed to Daniel Garrido-Sanz or Christoph Keel.

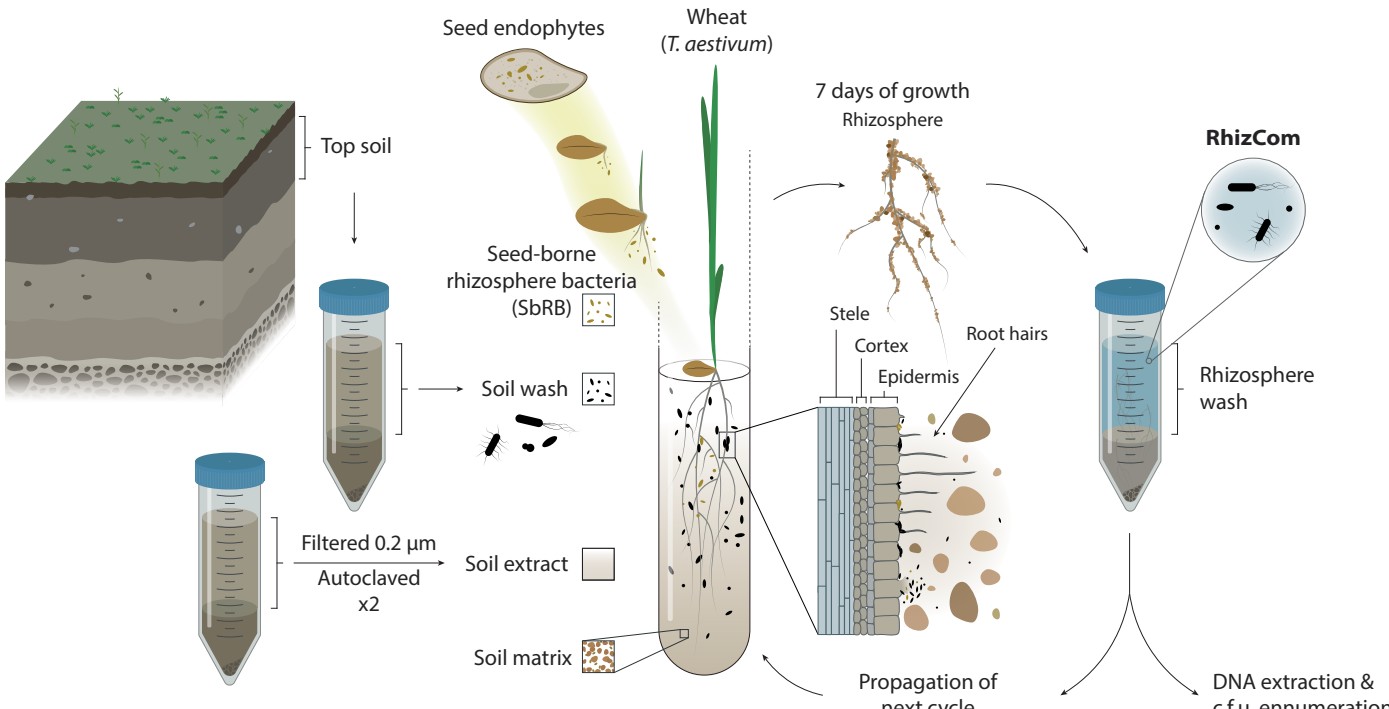

**Extended Data Fig. 1 | Schematic representation of the experimental design and microcosm used in this study.** The top bulk soil from a Swiss grassland bordering a wheat field was used to obtain a soil wash containing the soil microbial suspension, and a filtered and sterile soil extract to supplement the microcosm with soluble micronutrients and molecules present in the soil. Both the soil wash and the soil extract were introduced into sterile microcosms consisting of glass tubes containing a soil matrix (0.5-4 mm silt). A two-day-old wheat (*Triticum aestivum* var. Arina) seedling was then planted. After seven days of plant growth and microbiome development, the rhizospheres (roots and attached soil matrix) were collected, and the microbial cells were washed. The resulting rhizosphere wash was used to reinoculate the next propagation cycle, to extract total DNA for 16S rRNA amplicons and metagenome shotgun sequencing, and to enumerate c.f.u. of bacteria. The rhizosphere wash obtained at the end of the sixth propagation cycle contained the microbial community considered as the replicable rhizosphere community (RhizCom). Seed endophytes that were able to leave the seed tissues and establish in the wheat rhizosphere in the absence of the soil wash inoculum, were considered the seed-borne rhizosphere bacteria (SbRB).

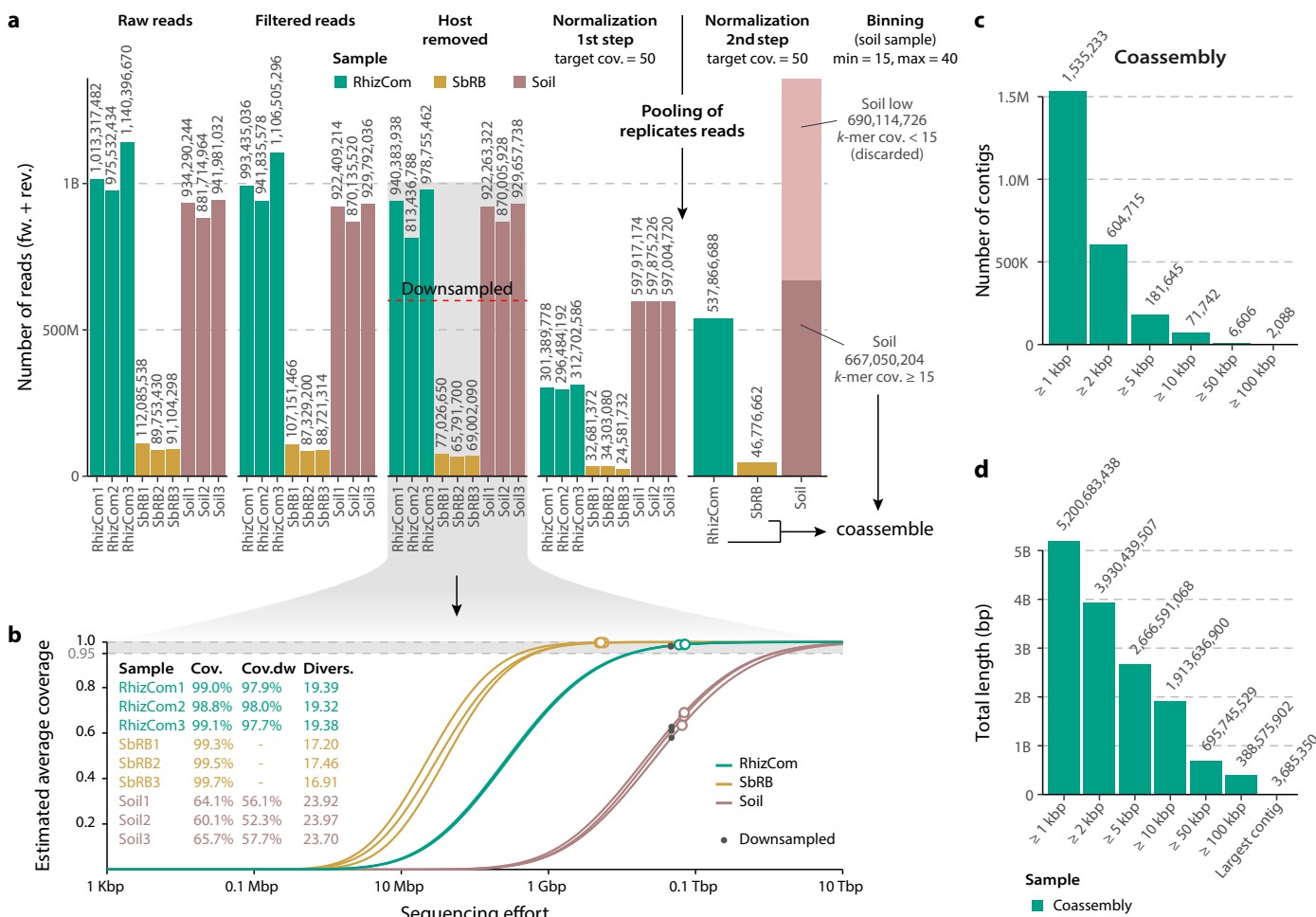

**Extended Data Fig. 2 | Processing of metagenome sequencing reads and coassembly statistics. a**, Combined number of forward and reverse reads obtained at each processing step in the RhizCom, SbRB and soil replicates. After host removal of host reads, reads were downsampled to a maximum of 600 million reads per sample (red dashed line) to account for computational limitations. **b**, Estimated sequence coverage per sample based on host-removed reads as a function of the sequencing effort. Empty dots along the curves indicate the average coverage achieved per replicate. Full black dots indicate coverage obtained after downsampling. Values of sample coverage (Cov.), sample coverage after downsampling (Cov.dw) and sample sequence diversity (Divers.) are indicated. **c,d**, Number of contigs (**c**), or total length in base pairs (bp) (**d**) after the coassembly of samples.

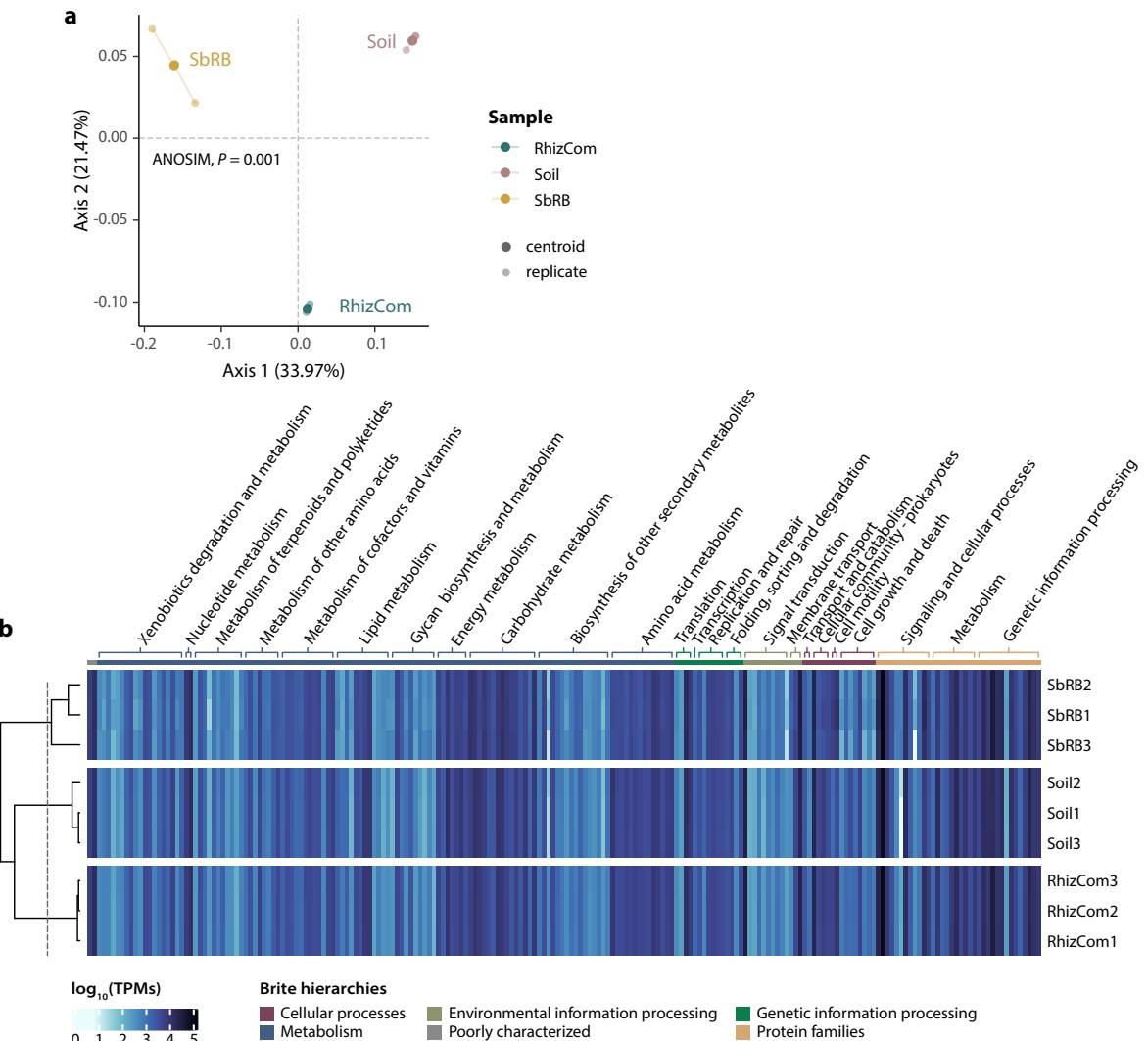

**Extended Data Fig. 3 | Overall functional content of samples. a**, Principal coordinate analysis (PCoA) of samples based on KEGGs TPM showing the two first axes. Statistical differences between samples were assessed based on Bray-Curtis dissimilarities using ANOSIM (Analysis of Similarities), using 9999 permutations. **b**, Distribution of KEGG BRITE hierarchies across samples. Samples were clustered based on the complete linkage of Euclidean distances. Only hierarchies with a total sum >1000 TPMs across samples were represented.

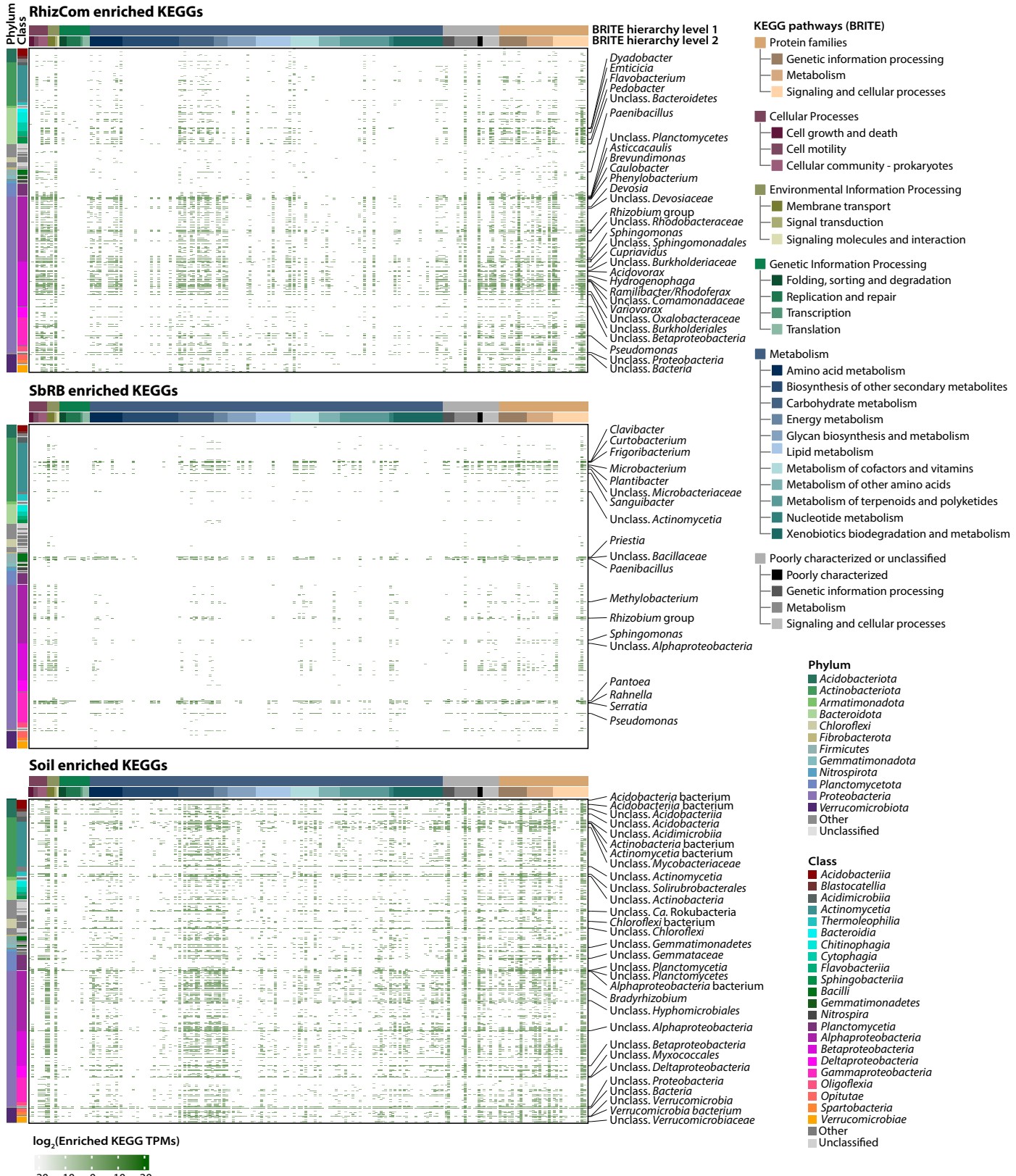

**Extended Data Fig. 4 | Taxonomic distribution of enriched KEGG annotations across communities.** Enriched KEGG annotations across communities based on DESeq2 differential abundance comparisons of TPMs. TPMs were summed per KEGG annotation and taxonomic rank at the genus level.

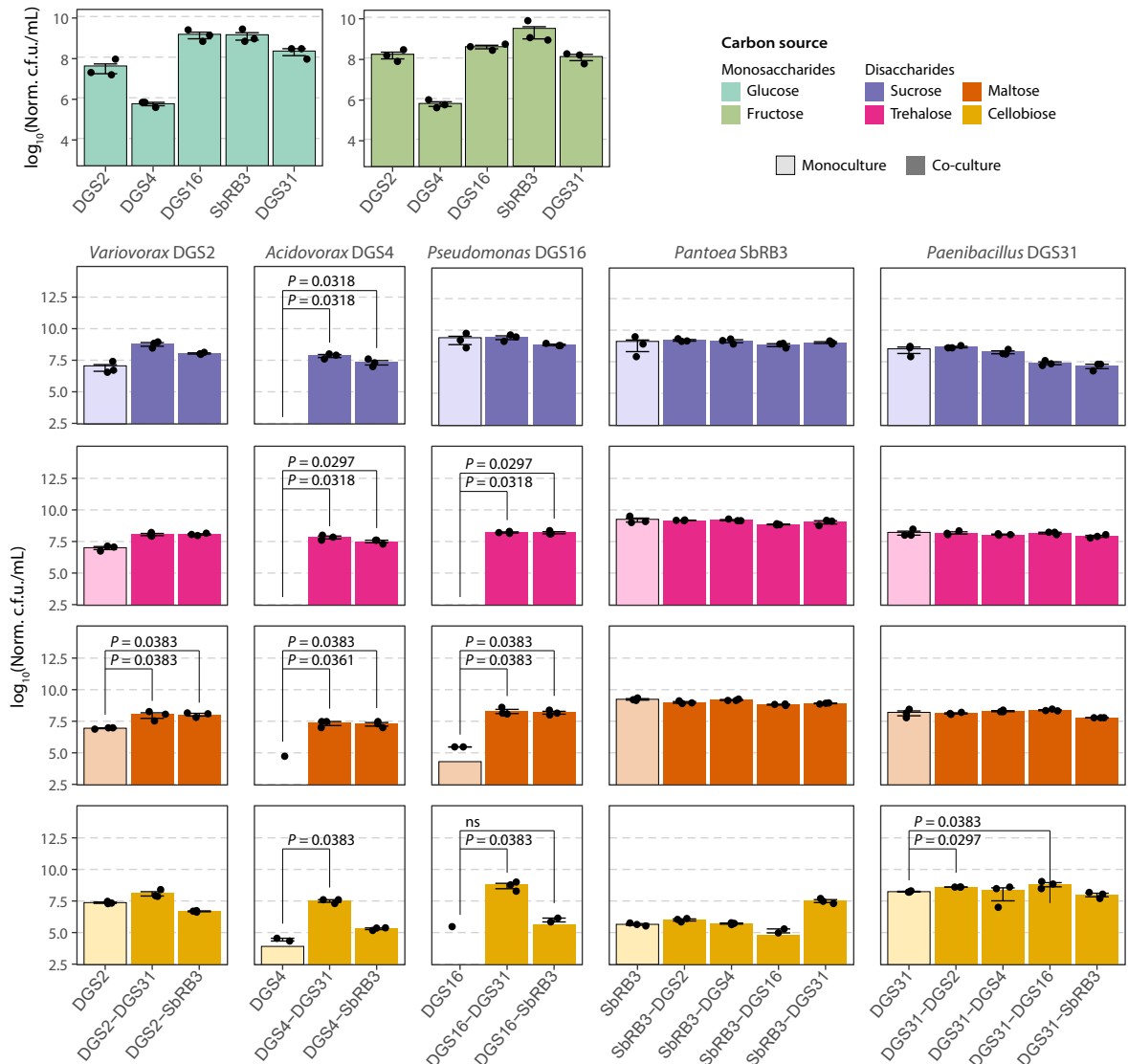

**Extended Data Fig. 5 | Disaccharide metabolism of seed-borne helper bacteria allows growth of partner strains.** Utilization of monosaccharides and disaccharides as sole carbon source by five bacterial strains isolated from the RhizCom, in monocultures (light bars) and co-cultures (dark bars). Bars represent mean normalized log10 c.f.u. after 24 h of growth in M9 medium supplemented with the six saccharides and vitamins (see Methods). Error bars represent the standard deviation and dots represent single biological replicates (n = 3). Normalization was calculated by subtracting the mean c.f.u. values of strains growing in M9 medium without any carbon source. Missing values result from negative values obtained after normalization. The ability of strains to achieve higher growth in co-cultures than in monocultures was tested for significance using the one-way Wilcoxon test with the "greater" alternative hypothesis. Only significant comparisons (P < 0.05) are reported. The seed-borne helper bacteria *Pantoea* and *Paenibacillus* support the growth of *Acidovorax* DGS4 and *Pseudomonas* DGS16 partners when grown with disaccharides as the sole carbon source. *Paenibacillus* also supports the growth of *Pantoea* when cellobiose is the sole carbon source.

# Reporting Summary

## Statistics

For all statistical analyses, confirm that the following items are present in the figure legend, table legend, main text, or Methods section.

| n/a | Confirmed | |
|---|---|---|
| ☐ | ☒ | The exact sample size (*n*) for each experimental group/condition, given as a discrete number and unit of measurement |
| ☐ | ☒ | A statement on whether measurements were taken from distinct samples or whether the same sample was measured repeatedly |
| ☐ | ☒ | The statistical test(s) used AND whether they are one- or two-sided *Only common tests should be described solely by name; describe more complex techniques in the Methods section.* |
| ☒ | ☐ | A description of all covariates tested |
| ☐ | ☒ | A description of any assumptions or corrections, such as tests of normality and adjustment for multiple comparisons |
| ☐ | ☒ | A full description of the statistical parameters including central tendency (e.g. means) or other basic estimates (e.g. regression coefficient) AND variation (e.g. standard deviation) or associated estimates of uncertainty (e.g. confidence intervals) |
| ☐ | ☒ | For null hypothesis testing, the test statistic (e.g. *F*, *t*, *r*) with confidence intervals, effect sizes, degrees of freedom and *P* value noted *Give P values as exact values whenever suitable.* |
| ☒ | ☐ | For Bayesian analysis, information on the choice of priors and Markov chain Monte Carlo settings |
| ☒ | ☐ | For hierarchical and complex designs, identification of the appropriate level for tests and full reporting of outcomes |
| ☒ | ☐ | Estimates of effect sizes (e.g. Cohen's *d*, Pearson's *r*), indicating how they were calculated |

*Our web collection on statistics for biologists contains articles on many of the points above.*

## Software and code

Policy information about availability of computer code

| Data collection | Software used: Microsoft Excel (version 2409 Build 16.0.18025.20160). |
|---|---|
| Data analysis | All the analyses were performed on a Dell workstation with an Intel(R) Xeon(R) Gold 6248R CPU processor, 384 GB of RAM, and Windows 10 Pro operating system.<br><br>Metagenome analyses were performed in a Linux environment, using Ubuntu 18.04 distribution, operating on the Windows Subsystem for Linux (version 2.3.26.0). The software used were fastp (version 0.23.2), bowtie2 (version 2.5.1), SAMtools (version 1.7), Nonpareil3 (version 3.401), BBnorm (version 37.62), Seqkit (version 2.2.0), MEGAHIT (version 1.2.9), QUAST (version 5.2.0), SqueezeMeta (version 1.6.2), diamond (version 2.0.15.153), Prodigal (v2.6.3), HMMER (v3.4), CONCOCT (version 1.1.0), MetaBAT 2 (version 1:2.15), MaxBin (version 2.2.6), DASTool (version 1.1.1), CheckM (version1.1.6), PhyloPhlAn (version 3.0.67) and RAxML (version 8.2.12)<br><br>Data processing, plotting and statistical analyses were performed using R (version 4.1.1 or 4.4.1) on RStudio (version 2024.04.2 Build 764). The packages used were agricolae (version 1.3-7), ape (version 5.8), Biostrings (version 2.70.3), car (version 3.1-3), circlize (version 0.4.16), ComplexHeatmap (version 2.20.0), dada2 (version 1.30.0), data.table (version 1.15.4), DECIPHER (version 2.30.0), DESeq2 (version 1.44.0), dplyr (version 1.1.4), ggalluvial (version 0.12.5), ggforce (0.4.2), ggh4x (version 0.2.8), ggnewscale (version 0.5.0), ggplot2 (version 3.5.1), ggplotify (version 0.1.2), ggpmisc (version 0.6.0), ggpubr (version 0.6.0), ggrepel (version 0.9.5), ggsci (version 3.2.0), ggtern (version 3.5.0), ggtree (version 3.12.0), ggtreeExtra (version 1.14.0), gridExtra (version 2.3), KEGGREST (version 1.44.1), metagenomeSeq (version 1.43.0), metagMisc (version 0.5.0), Nonpareil (version 3.5.3), pairwiseAdonis (version 0.4.1), pals (version 1.9), pathview (version 1.44.0), phangorn (version 2.11.1), phyloseq (version 1.46.0), phyloseq.extended (version 0.1.0.9000), picante (version 1.8.2), plotly (version 4.10.4), qiime2R (version 0.99.6), ranacapa (version0.1.0), RColorBrewer (version 1.1-3), scales (version 1.3.0), scico (version 1.5.0), seqinr (version 4.2-36), speedyseq (version 0.5.3.9021), SQMtools (version 1.6.3), stats (version 3.1-3) tidyverse (version 2.0.0) and vegan (version 2.6-6.1). |

Data used for statistical analyses were tested using the Saphiro-Wilk normality test, and the Levene's test for homogeneity of variances. Data that did not have a normal distribution or homogeneous variances were analyzed with the agricolae v1.4-5 R package, using the non-parametric Kruskal-Wallis rank sum test, with Fisher's least significant difference (LSD) post hoc criterium, and correction of P values using the false discovery rate (fdr). Pairwise comparisons were performed using two-sided Wilcoxon rank-sum tests within the stats v4.4.1 R package. For testing increases in bacterial growth when co-cultured with helper strains, one-sided Wilcoxon rank-sum tests were used, with the "greater" alternative hypothesis. Data that followed a normal distribution and with homogeneous variances were analyzed using PERMANOVA (permutational multivariate analyses of variance) or ANOSIM (analysis of similarity), using with the adonis2 or anosim functions, respectively, within the vegan v1.5-4 R package, and using 9999 permutations. Spearman correlations were calculated using the stat_cor function within the ggpubr R package, and data was fitted to a general additive model (GAM), using a $k = 3$ with the geom_smooth ggplot2 v3.5.1 R function, and a confidence interval of 0.95. No statistical method was used to predetermine sample size.

The code used for the analysis of the 16S rRNA gene amplicon data, metagenomic shotgun data, and other analyses reported in this study is available on GitHub (https://github.com/dgarrs/RhizCom) and Zenodo (https://doi.org/10.5281/zenodo.13969370).

For manuscripts utilizing custom algorithms or software that are central to the research but not yet described in published literature, software must be made available to editors and reviewers. We strongly encourage code deposition in a community repository (e.g. GitHub). See the Nature Portfolio guidelines for submitting code & software for further information.

# Data

Policy information about availability of data

All manuscripts must include a data availability statement. This statement should provide the following information, where applicable:
- Accession codes, unique identifiers, or web links for publicly available datasets
- A description of any restrictions on data availability
- For clinical datasets or third party data, please ensure that the statement adheres to our policy

Raw reads from both 16S rRNA gene amplicons and metagenomic shotgun have been deposited in the NCBI Sequence Read Archive (RSA) database and are publicly available under the BioProject accession number PRJNA1169405. The 16S rRNA gene sequences of the five isolated bacterial strains reported in this work have been submitted to the NCBI GenBank database under the following accession numbers: Variovorax sp. DGS2, PQ776219; Acidovorax sp. DGS4, PQ776220; Pseudomonas sp. DGS16, PQ776221; Paenibacillus sp. DGS31, PQ776222; and Pantoea sp. SbRB3, PQ776223. Other raw data generated in this study are provided in the Supplementary information or in the GitHub repository https://github.com/dgarrs/RhizCom, and available on Zenodo under https://doi.org/10.5281/zenodo.13969370.

Databases and datasets used in this study are available as follows. Wheat reference genome (Triticum aestivum, NCBI GenBank acc. no. GCA_903993985.1). SILVA database v138.1 (https://www.arb-silva.de/documentation/release-138.1/). Databases GeneBank nr, eggNOG, KEGG and Pfam were downloaded using the script make_databases.pl from SqueezeMeta. PhyloPhlAn database of 400 universal amino acid marker sequences was downloaded using the phylophlan_setup_database script of PhyloPhlAn. All databases were downloaded in May 2023.

# Research involving human participants, their data, or biological material

Policy information about studies with human participants or human data. See also policy information about sex, gender (identity/presentation), and sexual orientation and race, ethnicity and racism.

| | |
|---|---|
| Reporting on sex and gender | N/A. |
| Reporting on race, ethnicity, or other socially relevant groupings | N/A. |
| Population characteristics | N/A. |
| Recruitment | N/A. |
| Ethics oversight | N/A. |

Note that full information on the approval of the study protocol must also be provided in the manuscript.

# Field-specific reporting

Please select the one below that is the best fit for your research. If you are not sure, read the appropriate sections before making your selection.

☒ Life sciences ☐ Behavioural & social sciences ☐ Ecological, evolutionary & environmental sciences

For a reference copy of the document with all sections, see nature.com/documents/nr-reporting-summary-flat.pdf

# Life sciences study design

All studies must disclose on these points even when the disclosure is negative.

| | |
|---|---|
| Sample size | All sample sizes are described in the Method section of the paper. Microbiome (16S rRNA amplicon sequencing) samples were obtained as a pool of 4 individual plant rhizospheres. Per sample, 4 pools were obtained and sequenced. For metagenomes (shotgun illumina sequencing), samples were as above, but only 3 out of the 4 pools were sequenced. The choice of samples to be sequenced by shotgun was based on the most apparent similar ones in terms of amplicon profiling. The selected sample sizes provide sufficient number of replicates for meaningful ecological comparisons while balancing sequencing costs and data quality. |
| Data exclusions | No data were excluded for microbiome analysis (16S rRNA amplicon sequencing). For metagenomes (shotgun illumina sequencing), one out of the four replicate pools per sample were excluded based on the most apparent dissimilar sample in terms of amplicon profiling. |
| Replication | All samples contained at least 3 replicates, product of the pooling of four plant rhizospheres. All replication attempts were successful. |
| Randomization | N/A for this study. Sample collection and processing was driven by experimental design rather than random assignment. Microbiome and metagenome samples were obtained from plant rhizospheres, where pooling was necessary to capture biological variability. In addition, metagenome sequencing was performed on samples selected based on similarity of amplicon profiles to ensure representative shotgun sequencing. Given these constraints, randomization was neither practical nor relevant to the study objectives. |
| Blinding | Blinding was not possible for this study. The investigator that collected the data also performed the analyses. |

# Reporting for specific materials, systems and methods

We require information from authors about some types of materials, experimental systems and methods used in many studies. Here, indicate whether each material, system or method listed is relevant to your study. If you are not sure if a list item applies to your research, read the appropriate section before selecting a response.

## Materials & experimental systems

| n/a | Involved in the study |
|---|---|
| ☒ | ☐ Antibodies |
| ☒ | ☐ Eukaryotic cell lines |
| ☒ | ☐ Palaeontology and archaeology |
| ☒ | ☐ Animals and other organisms |
| ☒ | ☐ Clinical data |
| ☒ | ☐ Dual use research of concern |
| ☐ | ☒ Plants |

## Methods

| n/a | Involved in the study |
|---|---|
| ☒ | ☐ ChIP-seq |
| ☒ | ☐ Flow cytometry |
| ☒ | ☐ MRI-based neuroimaging |

## Plants

| | |
|---|---|
| Seed stocks | Non-treated Triticum aestivum var. Arina seeds (Lot number 111.1001) were obtained from Delley seeds and plants Ltd, Switzerland, on March 2021. |
| Novel plant genotypes | N/A. |
| Authentication | Autentification is carried out by seed bank supplier. The variety is registered in the National Swiss Catalog (CN 1981), and the List of Recommended Varieties (LR 1981) |

