## [Peer Review File · Nature Microbiology]

Seed-borne bacteria drive wheat rhizosphere microbiome assembly via niche partitioning and facilitation

Corresponding Author: Dr Daniel Garrido-Sanz

Version 0:

Reviewer comments:

Reviewer #1

(Remarks to the Author)

This paper investigates how heritable seed-borne bacteria (SbRB) influence the assembly of the wheat rhizosphere microbiome, focusing on the priority effects and community dynamics during early plant development. The authors used Sequential Succession experiment where they used a sequential propagation method to create a stable and reproducible wheat rhizosphere microbiome (RhizCom) and reported that community was dominated by Proteobacteria and Bacteroidota. The authors also reported that SbRB to the dominant contributors to the RhizCom due to priority effects, exploiting early-stage niches through saccharide metabolism and niche facilitation. What was interesting for me is that despite integration with soil bacteria, seed-derived microbes retained their metabolic roles, influencing overall community assembly. In summary, the study emphasizes the potential of SbRB as early microbial colonizers in shaping plant-associated microbiomes, advancing strategies for reproducible microbiome manipulation.

In my opinion, the topic of microbial inheritance and understanding how seed borne microorganisms in the assembly of the plant microbiome is of high importance. The other novelty I see is the experiment where the authors tried to show how SbRB behaved when they coalesced with the soil community. Overall the manuscript addresses an important gap in understanding plant-microbiome interactions and assembly.

The authors used six propagation cycles (which I find adequate), with 16 microcosms prepared per cycle containing a single wheat seedling, which means 16 plants were used per propagation cycle. What is not clear is why they pooled 4 plants to extract DNA for community analysis. Wasn't it better to sequence individual roots to account for the variability.

Currently, the explanation of SbRB is spread across the introduction and results. A concise, standalone definition in the introduction (e.g., in the paragraph where SbRB is first mentioned) would enhance reader understanding.

I also find it difficult to understand how they isolated SbRB. Was it seed germinated in microcosms without the soil inoculum? How many seed or plants were used to define the SbRB community.

I also don't understand fully how they traced back the SbRB to the RhizCom. Was it looking for presence absence or did they use a different method. If presence absence, there must have been some bacteria that was found in both. In which case, what is the authors explanation for using such a methods.

How many bacterial isolates were obtained to represent SbRB? How do these compare to the results from the amplicon sequencing? In terms of number and identity?

I see some information in Supplementary figure 3 and 4 but if I get it correctly, these were from amplicon and metagenomic sequencing?

I also find authors to use a lot of abbreviation SM, SE, RW, RE, SbRB, RhizCom, NatComs, STB etc. SE and SM SW and RW are specific to experimental conditions and not widely recognizable. Writing "soil extract" and "soil matrix" "soil wash" and "rhizosphere wash" could be written in full to enhance clarity.

Minor comments:

The introduction could benefit from focusing on the

Some figures are data-dense and might benefit from simplification or annotations to highlight key findings.

Ensure all supplementary materials referenced are accessible and clearly indexed.

Address potential biases in microbiome assembly due to the artificial setup of microcosms compared to natural conditions.

Some terms like "niche facilitation" could be briefly defined for readers outside the field.

Update references to include more recent studies where applicable especially on inheritance

(Remarks on code availability)

Reviewer #2

(Remarks to the Author)

Your paper, "Heritable seed-borne bacteria drive wheat rhizosphere microbiome assembly..." was very clear in its observations and conclusions. By using amplicon sequencing to follow bacterial communities over the course of several generations of rhizosphere transfer, you were able to show that soil inoculated wheat rhizospheres looked very much like the control rhizospheres which were grown without any external inoculum. For this section you have all the necessary details in the supplemental methods where few people will ever look, and one line buried in the methods, seemingly as an afterthought saying you had an uninoculated treatment. For most of the paper I was confused how the SbRB was identified or defined but I think it is critical to mention early on in the paper how and why you generated that "negative control" or SbRB. I think you should make this methodological detail clear in both the abstract and the introduction, clearly explaining that you had two different treatments in your experiment: serial transfer of rhizospheres from soil inoculated or uninoculated plants. Also very important to mention both in the introduction and methods (but not necessarily the abstract) is that you grew the plant on a heat sterilized substrate and that all the plants came from surface sterilized seed, where you eliminated the influence of any seed surface microbiomes. One of the main reasons scientists have for so long thought that the plant microbiome derives from the soil, is that they never include negative controls (sterile soil and sterile inoculum) in their experiments and you should make clearer that you didn't continue making that mistake.

The metagenomics section was extensive, thorough and convincing in support of your hypothesis that SbRB are able to dominate the wheat rhizosphere thanks to their abilities to degrade specific root secreted saccharides. My only comment here is to ask if you could include a specific look at bacterial functions which may be beneficial to plants regardless of their enrichment, for example genes involved in nitrogen fixation, hormone production, siderophores, antibiotic production (ie. phenazines), etc. This might be a subfigure in your Figure 3, perhaps replacing 3C which seems unnecessary. Currently the only functions you seemed to look at were the most abundant/enriched, while it would be very interesting to many readers such as myself to know whether seed transmitted bacteria already bring with them most of the functions that wheat plants might desire to have in their rhizospheres.

In your discussion, you do a great job going over your results, but you haven't done much to contextualize your experiment in relation to other plants or what is already known about microbiome transmission. Were your wheat rhizobiomes similar to those observed in other published studies? Do you think that wheat rhizobiomes form in the same way that all other plants do (ie. other monocots like maize or dicots like Arabidopsis and soybean)? Why did you choose to work with wheat versus Arabidopsis for example? How do seed transmitted bacteria exit the endosphere and colonize the rhizosphere? Have you heard of the rhizophagy cycle and are most of these bacteria participants in that process? Were any of the rhizobacteria you observed, part of the seed transmitted core microbiomes that are described in your references 28 and 29? It may be useful to discuss whether there were any limitations to your study, for example, might there have been some sort of artifact in microbiome transmission created by sterilizing the seed surface before planting?

(Remarks on code availability)

Reviewer #3

(Remarks to the Author)

The authors suggest a sequential propagation strategy to generate a complex and reproducible plant rhizosphere community (RhizCom) to analyse plant microbiome assembly in wheat. Seed-borne rhizosphere bacteria emerged as the dominant microbiome source because of their ability to degrade specific saccharides and niche facilitation. The strength of the manuscript is the suggestion of the reproducible RhizCom to figure out the core of transmitted bacteria and their metabolic pattern (assimilation of disaccharides). However, I would suggest to consider the following comments and questions to improve the manuscript:

Abstract

Please exactly justify the statement in what: Our results advance our understanding of the principles governing microbial community dynamics in early plant development.

Unusual abbreviations, e.g. seed-borne rhizosphere bacteria (SbRB) seed-transmitted bacteria (STB) make it difficult to read. Are all these bacteria endophytes? – this is crucial to show.

Add a few more details about the 821 metagenome-assembled genomes.

Knowledge gap

Line 48: surprising, as soil and rhizosphere microbiomes are among the most diverse environments on Earth – this is true for the soil microbiome but not for the rhizosphere!

Lines 48-57: There are not only these two approaches to get mechanistic insights! Here you can find examples for this: Matsumoto H, Fan X, Wang Y, Kusstatscher P, Duan J, Wu S, Chen S, Qiao K, Wang Y, Ma B, Zhu G, Hashidoko Y, Berg G, Cernava T, Wang M. Bacterial seed endophyte shapes disease resistance in rice. *Nat Plants*. 2021 Jan;7(1):60-72. A Bergna, T Cernava, M Rändler, R Grosch, C Zachow, G Berg. Tomato seeds preferably transmit plant beneficial endophytes. *Phytobiomes Journal* 2 (4), 183-193

Line 64: emphasized that seed-transmitted bacteria (STB) are also an important source of the rhizobiome.

Line 70: However, how both SbRB and soil microbiomes coalesce into the plant rhizobiome

and whether specific metabolic signatures exist within the heritable plant microbial diversity remains poorly investigated. This paragraph needs more details – here, you should summarize current results and identify a clear knowledge gap. Mention research hypothesis would help as well.

Experimental design

The study focusses on bacteria only but also fungi are transmitted by seeds and wheat is ubiquitously mycorrhizal.

Rhizobiome was frequently used – how is it defined? Ref?

Plant microbiome assembly strongly depends on the species and even on the cultivar. What was the reason to select this one?

To include more species or cultivars would allow more general answers and show that Fig 6 summarize a general picture.

See Ref: Michl K, Berg G, Cernava T. The microbiome of cereal plants: The current state of knowledge and the potential for future applications. *Environ Microbiome*. 2023 Mar 31;18(1):28.

Where the „successive cycles of microbiome propagation“ published before or is it your development? This is important for novelty of the study.

Results

Line 157: the soil was enriched in functions – soil is more a seed and storage bank for bacteria, I don't think that enrich is the right word here

Line 203-204: nicotinate, and nicotinamide metabolism and riboflavin metabolism – what does this mean?

Line 2021: hat SbRB initiate the assimilation of disaccharides in the wheat rhizosphere, which can then be utilized by a larger number of RhizCom members.

I mean this does not explain the selective enrichment in the rhizosphere because all bacteria would love that?

Please show the overlap of amplicons and MAGs in more detail.

Fig 6 try to integrate more details about the bacterial structure and function (Remarks on code availability)

ok

Decision Letter:

10th December 2024

Dear Dr Garrido-Sanz,

Thank you for your patience while your manuscript "Heritable seed-borne bacteria drive wheat rhizosphere microbiome assembly through priority effects during sequential succession" was under peer-review at Nature Microbiology. It has now been seen by 3 referees, whose expertise and comments you will find at the of this email. You will see from their comments below that while they find your work of interest, some important points are raised, though I'm pleased to share that these concerns seem relatively minor and should be straightforward to address. We are very interested in the possibility of publishing your study in Nature Microbiology, however we would like to consider your response to these concerns in the form of a revised manuscript before we make a final decision on publication.

If you have not done so already please begin to revise your manuscript so that it conforms to our Article format instructions at <http://www.nature.com/nmicrobiol/info/final-submission/>

The usual length limit for a Nature Microbiology Article is six display items (figures or tables) and 3,000 words. We have some flexibility, and can allow a revised manuscript at 3,500 words, but please consider this a firm upper limit. There is a trade-off of ~250 words per display item, so if you need more space, you could move a Figure or Table to Supplementary Information.

Some reduction could be achieved by focusing any introductory material and moving it to the start of your opening 'bold' paragraph, whose function is to outline the background to your work, describe in a sentence your new observations, and explain your main conclusions. The discussion should also be limited. Methods should be described in a separate section following the discussion, we do not place a word limit on Methods.

Nature Microbiology titles should give a sense of the main new findings of a manuscript, and should not contain punctuation. Please keep in mind that we strongly discourage active verbs in titles, and that they should ideally fit within 90 characters each (including spaces).

We strongly support public availability of data. Please place the data used in your paper into a public data repository, if one exists, or alternatively, present the data as Source Data or Supplementary Information. If data can only be shared on request, please explain why in your Data Availability Statement, and also in the correspondence with your editor. For some data types, deposition in a public repository is mandatory - more information on our data deposition policies and available repositories can

be found at <https://www.nature.com/nature-research/editorial-policies/reporting-standards#availability-of-data>.

Please include a data availability statement as a separate section after Methods but before references, under the heading "Data Availability". This section should inform readers about the availability of the data used to support the conclusions of your study. This information includes accession codes to public repositories (data banks for protein, DNA or RNA sequences, microarray, proteomics data etc...), references to source data published alongside the paper, unique identifiers such as URLs to data repository entries, or data set DOIs, and any other statement about data availability. At a minimum, you should include the following statement: "The data that support the findings of this study are available from the corresponding author upon request", mentioning any restrictions on availability. If DOIs are provided, we also strongly encourage including these in the Reference list (authors, title, publisher (repository name), identifier, year). For more guidance on how to write this section please see: <http://www.nature.com/authors/policies/data/data-availability-statements-data-citations.pdf>

To improve the accessibility of your paper to readers from other research areas, please pay particular attention to the wording of the paper's opening bold paragraph, which serves both as an introduction and as a brief, non-technical summary in about 150 words. If, however, you require one or two extra sentences to explain your work clearly, please include them even if the paragraph is over-length as a result. The opening paragraph should not contain references. Because scientists from other sub-disciplines will be interested in your results and their implications, it is important to explain essential but specialised terms concisely. We suggest you show your summary paragraph to colleagues in other fields to uncover any problematic concepts.

If your paper is accepted for publication, we will edit your display items electronically so they conform to our house style and will reproduce clearly in print. If necessary, we will re-size figures to fit single or double column width. If your figures contain several parts, the parts should form a neat rectangle when assembled. Choosing the right electronic format at this stage will speed up the processing of your paper and give the best possible results in print. We would like the figures to be supplied as vector files - EPS, PDF, AI or postscript (PS) file formats (not raster or bitmap files), preferably generated with vector-graphics software (Adobe Illustrator for example). Please try to ensure that all figures are non-flattened and fully editable. All images should be at least 300 dpi resolution (when figures are scaled to approximately the size that they are to be printed at) and in RGB colour format. Please do not submit Jpeg or flattened TIFF files. Please see also 'Guidelines for Electronic Submission of Figures' at the end of this letter for further detail.

Figure legends must provide a brief description of the figure and the symbols used, within 350 words, including definitions of any error bars employed in the figures.

When submitting the revised version of your manuscript, please pay close attention to our [href="https://www.nature.com/nature-research/editorial-policies/image-integrity">Digital Image Integrity Guidelines.](https://www.nature.com/nature-research/editorial-policies/image-integrity) and to the following points below:

Please include a statement before the acknowledgements naming the author to whom correspondence and requests for materials should be addressed.

Finally, we require authors to include a statement of their individual contributions to the paper -- such as experimental work, project planning, data analysis, etc. -- immediately after the acknowledgements. The statement should be short, and refer to authors by their initials. For details please see the Authorship section of our joint Editorial policies at http://www.nature.com/authors/editorial_policies/authorship.html

* include a point-by-point response to any editorial suggestions and to our referees. Please include your response to the editorial suggestions in your cover letter, and please upload your response to the referees as a separate document.

* ensure it complies with our format requirements for Letters as set out in our guide to authors at www.nature.com/nmicrobiol/info/gta/

* state in a cover note the length of the text, methods and legends; the number of references; number and estimated final size of figures and tables

* resubmit electronically if possible using the link below to access your home page:

Link Redacted

*This url links to your confidential homepage and associated information about manuscripts you may have submitted or be reviewing for us. If you wish to forward this e-mail to co-authors, please delete this link to your homepage first.

Please ensure that all correspondence is marked with your Nature Microbiology reference number in the subject line.

Nature Microbiology is committed to improving transparency in authorship. As part of our efforts in this direction, we are now requesting that all authors identified as 'corresponding author' on published papers create and link their Open Researcher and Contributor Identifier (ORCID) with their account on the Manuscript Tracking System (MTS), prior to acceptance. This applies to primary research papers only. ORCID helps the scientific community achieve unambiguous attribution of all scholarly contributions. You can create and link your ORCID from the home page of the MTS by clicking on 'Modify my Springer Nature account'. For more information please visit www.springernature.com/orcid.

We hope to receive your revised paper within three weeks. If you cannot send it within this time, please let us know.

Yours sincerely,

Reviewer Expertise:

Referee #1: rhizosphere, ecology, metagenomics
Referee #2: plant microbiome, seed microbiome, 'omics
Referee #3: plant microbiome

Reviewers Comments:

Reviewer #1 (Remarks to the Author):

This paper investigates how heritable seed-borne bacteria (SbRB) influence the assembly of the wheat rhizosphere microbiome, focusing on the priority effects and community dynamics during early plant development. The authors used Sequential Succession experiment where they used a sequential propagation method to create a stable and reproducible wheat rhizosphere microbiome (RhizCom) and reported that community was dominated by Proteobacteria and Bacteroidota. The authors also reported that SbRB to the dominant contributors to the RhizCom due to priority effects, exploiting early-stage niches through saccharide metabolism and niche facilitation. What was interesting for me is that despite integration with soil bacteria, seed-derived microbes retained their metabolic roles, influencing overall community assembly. In summary, the study emphasizes the potential of SbRB as early microbial colonizers in shaping plant-associated microbiomes, advancing strategies for reproducible microbiome manipulation.

In my opinion, the topic of microbial inheritance and understanding how seed borne microorganisms in the assembly of the plant microbiome is of high importance. The other novelty I see is the experiment where the authors tried to show how SbRB behaved when they coalesced with the soil community. Overall the manuscript addresses an important gap in understanding plant-microbiome interactions and assembly.

The authors used six propagation cycles (which I find adequate), with 16 microcosms prepared per cycle containing a single wheat seedling, which means 16 plants were used per propagation cycle. What is not clear is why they pooled 4 plants to extract DNA for community analysis. Wasn't it better to sequence individual roots to account for the variability.

Currently, the explanation of SbRB is spread across the introduction and results. A concise, standalone definition in the introduction (e.g., in the paragraph where SbRB is first mentioned) would enhance reader understanding.

I also find it difficult to understand how they isolated SbRB. Was it seed germinated in microcosms without the soil inoculum? How many seed or plants were used to define the SbRB community.

I also don't understand fully how they traced back the SbRB to the RhizCom. Was it looking for presence absence or did they use a different method. If presence absence, there must have been some bacteria that was found in both. In which case, what is the authors explanation for using such a methods.

How many bacterial isolates were obtained to represent SbRB? How do these compare to the results from the amplicon sequencing? In terms of number and identity?

I see some information in Supplementary figure 3 and 4 but if I get it correctly, these were from amplicon and metagenomic sequencing?

I also find authors to use a lot of abbreviation SM, SE, RW, RE, SbRB, RhizCom, NatComs, STB etc. SE and SM SW and RW are specific to experimental conditions and not widely recognizable. Writing "soil extract" and "soil matrix" "soil wash" and "rhizosphere wash" could be written in full to enhance clarity.

Minor comments:

The introduction could benefit from focusing on the

Some figures are data-dense and might benefit from simplification or annotations to highlight key findings.

Ensure all supplementary materials referenced are accessible and clearly indexed.

Address potential biases in microbiome assembly due to the artificial setup of microcosms compared to natural conditions.

Some terms like "niche facilitation" could be briefly defined for readers outside the field.

Update references to include more recent studies where applicable especially on inheritance

Reviewer #2 (Remarks to the Author):

Your paper, "Heritable seed-borne bacteria drive wheat rhizosphere microbiome assembly..." was very clear in its observations and conclusions. By using amplicon sequencing to follow bacterial communities over the course of several generations of rhizosphere transfer, you were able to show that soil inoculated wheat rhizospheres looked very much like the control rhizospheres which were grown without any external inoculum. For this section you have all the necessary details in the supplemental methods where few people will ever look, and one line buried in the methods, seemingly as an afterthought saying you had an uninoculated treatment. For most of the paper I was confused how the SbRB was identified or defined but I think it is critical to mention early on in the paper how and why you generated that "negative control" or SbRB. I think you should make this methodological detail clear in both the abstract and the introduction, clearly explaining that you had two different treatments in your experiment: serial transfer of rhizospheres from soil inoculated or uninoculated plants. Also very important to mention both in the introduction and methods (but not necessarily the abstract) is that you grew the plant on a heat sterilized substrate and that all the plants came from surface sterilized seed, where you eliminated the influence of any seed surface microbiomes. One of the main reasons scientists have for so long thought that the plant microbiome derives from the soil, is that they never include negative controls (sterile soil and sterile inoculum) in their experiments and you should make clearer that you didn't continue making that mistake.

The metagenomics section was extensive, thorough and convincing in support of your hypothesis that SbRB are able to dominate the wheat rhizosphere thanks to their abilities to degrade specific root secreted saccharides. My only comment here is to ask if you could include a specific look at bacterial functions which may be beneficial to plants regardless of their enrichment, for example genes involved in nitrogen fixation, hormone production, siderophores, antibiotic production (ie. phenazines), etc. This might be a subfigure in your Figure 3, perhaps replacing 3C which seems unnecessary. Currently the only functions you seemed to look at were the most abundant/enriched, while it would be very interesting to many readers such as myself to know whether seed transmitted bacteria already bring with them most of the functions that wheat plants might desire to have in their rhizospheres.

In your discussion, you do a great job going over your results, but you haven't done much to contextualize your experiment in relation to other plants or what is already known about microbiome transmission. Were your wheat rhizobiomes similar to those observed in other published studies? Do you think that wheat rhizobiomes form in the same way that all other plants do (ie. other monocots like maize or dicots like Arabidopsis and soybean)? Why did you choose to work with wheat versus Arabidopsis for example? How do seed transmitted bacteria exit the endosphere and colonize the rhizosphere? Have you heard of the rhizophagy cycle and are most of these bacteria participants in that process? Were any of the rhizobacteria you observed, part of the seed transmitted core microbiomes that are described in your references 28 and 29? It may be useful to discuss whether there were any limitations to your study, for example, might there have been some sort of artifact in microbiome transmission created by sterilizing the seed surface before planting?

Reviewer #3 (Remarks to the Author):

The authors suggest a sequential propagation strategy to generate a complex and reproducible plant rhizosphere community (RhizCom) to analyse plant microbiome assembly in wheat. Seed-borne rhizosphere bacteria emerged as the dominant microbiome source because of their ability to degrade specific saccharides and niche facilitation. The strength of the manuscript is the suggestion of the reproducible RhizCom to figure out the core of transmitted bacteria and their metabolic pattern (assimilation of disaccharides). However, I would suggest to consider the following comments and questions to improve the manuscript:

Abstract

Please exactly justify the statement in what: Our results advance our understanding of the principles governing microbial community dynamics in early plant development.

Unusual abbreviations, e.g. seed-borne rhizosphere bacteria (SbRB) seed-transmitted bacteria (STB) make it difficult to read. Are all these bacteria endophytes? – this is crucial to show.

Add a few more details about the 821 metagenome-assembled genomes.

Knowledge gap

Line 48: surprising, as soil and rhizosphere microbiomes are among the most diverse environments on Earth – this is true for the soil microbiome but not for the rhizosphere!

Lines 48-57: There are not only these two approaches to get mechanistic insights! Here you can find examples for this: Matsumoto H, Fan X, Wang Y, Kusstatscher P, Duan J, Wu S, Chen S, Qiao K, Wang Y, Ma B, Zhu G, Hashidoko Y, Berg G, Cernava T, Wang M. Bacterial seed endophyte shapes disease resistance in rice. *Nat Plants*. 2021 Jan;7(1):60-72.
A Bergna, T Cernava, M Rändler, R Grosch, C Zachow, G Berg. Tomato seeds preferably transmit plant beneficial endophytes. *Phytobiomes Journal* 2 (4), 183-193

Line 64: emphasized that seed-transmitted bacteria (STB) are also an important source of the rhizobiome.

Line 70: However, how both SbRB and soil microbiomes coalesce into the plant rhizobiome and whether specific metabolic signatures exist within the heritable plant microbial diversity remains poorly investigated.

This paragraph needs more details – here, you should summarize current results and identify a clear knowledge gap. Mention research hypothesis would help as well.

Experimental design

The study focusses on bacteria only but also fungi are transmitted by seeds and wheat is ubiquitously mycorrhizal.

Rhizobiome was frequently used – how is it defined? Ref?

Plant microbiome assembly strongly depends on the species and even on the cultivar. What was the reason to select this one?

To include more species or cultivars would allow more general answers and show that Fig 6 summarize a general picture.

See Ref: Michl K, Berg G, Cernava T. The microbiome of cereal plants: The current state of knowledge and the potential for future applications. Environ Microbiome. 2023 Mar 31;18(1):28.

Where the „successive cycles of microbiome propagation“ published before or is it your development? This is important for novelty of the study.

Results

Line 157: the soil was enriched in functions – soil is more a seed and storage bank for bacteria, I don't think that enrich is the right word here

Line 203-204: nicotinate, and nicotinamide metabolism and riboflavin metabolism – what does this mean?

Line 2021: hat SbRB initiate the assimilation of disaccharides in the wheat rhizosphere, which can then be utilized by a larger number of RhizCom members.

I mean this does not explain the selective enrichment in the rhizosphere because all bacteria would love that?

Please show the overlap of amplicons and MAGs in more detail.

Fig 6 try to integrate more details about the bacterial structure and function

Reviewer #3 (Remarks on code availability):

ok

Version 1:

Reviewer comments:

Reviewer #1

(Remarks to the Author)

Thank you for your detailed responses and revisions. Overall, I am pleased with the improvements made to the manuscript. I appreciate your thoughtful consideration of my comments and the effort to address them.

Best regards,

Reviewer #1

(Remarks on code availability)

Reviewer #2

(Remarks to the Author)

Thank you for attempting to address my questions and comments. My most important requests were addressed, while a few less important ones were not possible due to length limitations. It is difficult to include so much information in a short, condensed publication and I applaud you for doing so. Try look a little more at rhizophagy in the future though. ;-)

(Remarks on code availability)

Reviewer #3

(Remarks to the Author)

The revision is really well done, and I am fully satisfied with the current manuscript. Congratulations!

Only three minor things:

Line 59: typing error, remove one comma

Line 63-64: "major source" for crops I would agree but for native plants, according to the current knowledge, the proportion of bacteria originating from seed or soil strongly depends on plant species. So may be rephrase a bit.

And only one question of personal interest: Pantoea is more and more considered as opportunistic plant and human pathogens. F. e. EFSA has forbidden any application as plant growth promoting bacteria. Could you find any pathogenicity factor in your strains/MAGs?

(Remarks on code availability)

Decision Letter:

Our ref: NMICROBIOL-24103250A

7th February 2025

Dear Dr. Garrido-Sanz,

Thank you for submitting your revised manuscript "Seed-borne bacteria drive wheat rhizosphere microbiome assembly through priority effects" (NMICROBIOL-24103250A). It has now been seen by the original referees and their comments are below. The reviewers find that the paper has improved in revision, and therefore we'll be happy in principle to publish it in Nature Microbiology, pending minor revisions to satisfy the referees' final requests and to comply with our editorial and formatting guidelines.

Thank you again for your interest in Nature Microbiology Please do not hesitate to contact me if you have any questions.

Sincerely,

Reviewer #1 (Remarks to the Author):

Thank you for your detailed responses and revisions. Overall, I am pleased with the improvements made to the manuscript. I appreciate your thoughtful consideration of my comments and the effort to address them.

Best regards,

Reviewer #1

Reviewer #2 (Remarks to the Author):

Thank you for attempting to address my questions and comments. My most important requests were addressed, while a few less important ones were not possible due to length limitations. It is difficult to include so much information in a short, condensed publication and I applaud you for doing so. Try look a little more at rhizophagy in the future though. ;-)

Reviewer #3 (Remarks to the Author):

The revision is really well done, and I am fully satisfied with the current manuscript. Congratulations!

Only three minor things:

Line 59: typing error, remove one comma

Line 63-64: "major source" for crops I would agree but for native plants, according to the current knowledge, the proportion of bacteria originating from seed or soil strongly depends on plant species. So may be rephrase a bit.

And only one question of personal interest: Pantoea is more and more considered as opportunistic plant and human pathogens. F. e. EFSA has forbidden any application as plant growth promoting bacteria. Could you find any pathogenicity factor in your strains/MAGs?

Version 2:

Decision Letter:

26th February 2025

Dear Dr Garrido-Sanz,

I am pleased to accept your Article "Seed-borne bacteria drive wheat rhizosphere microbiome assembly via niche partitioning and facilitation" for publication in Nature Microbiology. Thank you for having chosen to submit your work to us and many congratulations.

Authors may need to take specific actions to achieve [compliance](https://www.springernature.com/gp/open-research/funding/policy-compliance-faqs) with funder and institutional open access mandates. If your research is supported by a funder that requires immediate open access (e.g. according to [Plan S principles](https://www.springernature.com/gp/open-research/plan-s-compliance)) then you should select the gold OA route, and we will direct you to the compliant route where possible. For authors selecting the subscription publication route, the journal's standard licensing terms will need to be accepted, including [self-archiving policies](https://www.nature.com/nature-portfolio/editorial-policies/self-archiving-and-license-to-publish). Those licensing terms will supersede any other terms that the author or any third party may assert apply to any version of the manuscript.

With kind regards,

P.S. Click on the following link if you would like to recommend Nature Microbiology to your librarian
<http://www.nature.com/subscriptions/recommend.html#forms>

** Visit the Springer Nature Editorial and Publishing website at http://editorial-jobs.springernature.com?utm_source=ejP_NMicro_email&utm_medium=ejP_NMicro_email&utm_campaign=ejp_NMicro for more information about our career opportunities. If you have any questions please click [here](mailto:editorial.publishing.jobs@springernature.com).**

Point-by-point response to reviewers

Reviewer #1 (Remarks to the Author):

This paper investigates how heritable seed-borne bacteria (SbRB) influence the assembly of the wheat rhizosphere microbiome, focusing on the priority effects and community dynamics during early plant development. The authors used Sequential Succession experiment where they used a sequential propagation method to create a stable and reproducible wheat rhizosphere microbiome (RhizCom) and reported that community was dominated by Proteobacteria and Bacteroidota. The authors also reported that SbRB to the dominant contributors to the RhizCom due to priority effects, exploiting early-stage niches through saccharide metabolism and niche facilitation. What was interesting for me is that despite integration with soil bacteria, seed-derived microbes retained their metabolic roles, influencing overall community assembly. In summary, the study emphasizes the potential of SbRB as early microbial colonizers in shaping plant-associated microbiomes, advancing strategies for reproducible microbiome manipulation.

In my opinion, the topic of microbial inheritance and understanding how seed-borne microorganisms in the assembly of the plant microbiome is of high importance. The other novelty I see is the experiment where the authors tried to show how SbRB behaved when they coalesced with the soil community. Overall, the manuscript addresses an important gap in understanding plant-microbiome interactions and assembly.

We thank Reviewer #1 for taking the time to evaluate our work and for the thoughtful questions and constructive comments that helped us improve our manuscript. We are pleased that the importance of our work has been recognized.

Please note that all line number references in our responses correspond to the revised version of the manuscript. Given the strict text limitations imposed by the journal, we have addressed the reviewer's concerns as concisely as possible, while making every effort to incorporate all suggestions by adjusting and streamlining other sections of the manuscript.

The authors used six propagation cycles (which I find adequate), with 16 microcosms prepared per cycle containing a single wheat seedling, which means 16 plants were used per propagation cycle. What is not clear is why they pooled 4 plants to extract DNA for community analysis. Wasn't it better to sequence individual roots to account for the variability.

We thank the reviewer for this question. The decision to pool four plant rhizospheres for DNA extraction for community analysis and metagenomics was based on two main considerations:

1. Technically, the DNA yield from a single plant rhizosphere is typically very low. In our previous analysis, we observed that the yield from a single rhizosphere was often at the threshold required for successful amplification by PCR. This is also due to the presence of alcohols, polyphenols and other contaminants in samples extracted from the rhizosphere, which particularly affect PCR efficiency. Pooling the rhizospheres of four plants increases the total DNA yield and allows dilution to the normalized 10 ng per sample for each reaction. This dilution reduces the effect of contaminants and improves PCR performance.
2. We acknowledge that individual plant variability influences microbiome assembly and that this variability is significant, as demonstrated by our results on SbRB heterogeneity. However, in our experimental design, rhizosphere washes from 16 individual plants were combined at the end of each cycle to inoculate the following cycle. Over successive cycles, this approach reduces individual rhizobiome variability

and results in the stable RhizCom. A single-plant sequential inoculation strategy, transferring the rhizosphere community from one individual plant to another, would likely have resulted in a similar outcome due to the inherent heterogeneity within each seed-transmitted bacterial community that accumulates over sequential cycles.

We hope that these explanations adequately address the concerns raised by Reviewer #1.

Currently, the explanation of SbRB is spread across the introduction and results. A concise, standalone definition in the introduction (e.g., in the paragraph where SbRB is first mentioned) would enhance reader understanding.

We thank the reviewer for this suggestion. We have taken this comment into account and revised the part of the Introduction that deals with SbRB and STB. We realized that the use of seed-transmitted bacteria (STB) and seed-borne rhizosphere bacteria (SbRB) may have caused some confusion. We have reworded this paragraph (L62-L67) and removed the abbreviation STB for clarity. We believe the focus and definition of SbRB is now clearer.

I also find it difficult to understand how they isolated SbRB. Was it seed germinated in microcosms without the soil inoculum? How many seed or plants were used to define the SbRB community.

We apologize for not making this clearer in the first version of our manuscript. As Reviewer #1 understands correctly, the SbRB community was obtained from surface-sterilized seeds germinated in microcosms without soil inoculum for one cycle (i.e., 7 days of plant growth). To define the SbRB community, we used a total of 16 plants, pooled by 4. This is the same number of plants used in each RhizCom cycle. We have reworded the Methods section to make it clearer, which now reads: “*To recover the seed-borne rhizosphere bacteria (SbRB) community, 16 microcosms were set up for one cycle (seven days) of plant growth without addition of soil inoculum. After seven days, the rhizospheres were collected, pooled by 4, and processed as described above.*” (L346-L349).

In addition, and in response to the concerns of the other reviewers, this has been further emphasized throughout the manuscript. Specifically, we have introduced the following sentences:

1. Results: “*The SbRB community was obtained from one cycle of 16 uninoculated plants*”, L90.
2. Discussion: “*Our microcosms may introduce biases compared to natural conditions, such as the lack of soil heterogeneity and the use of surface-disinfected seeds, which in nature carry bacteria that may contribute to rhizobiome assembly*”, L259-L261.

I also don't understand fully how they traced back the SbRB to the RhizCom. Was it looking for presence absence or did they use a different method. If presence absence, there must have been some bacteria that was found in both. In which case, what is the authors explanation for using such a methods.

We apologize again for the lack of clarity in the first version of the manuscript. Since the only possible origin of RhizCom bacteria is either the soil inoculum or the SbRB, we looked for the presence of RhizCom ASVs in the source SbRB or soil communities to trace their origin (== exact ASV match). This has been further clarified in the Results section, which now reads: “*We examined the origin of RhizCom members by identifying exact matches of ASVs from either the SbRB or soil communities, as these are the only possible sources. Only a small number of RhizCom ASVs could be traced to either source community*” (L109-L111).

Indeed, there were a total of 5 RhizCom ASVs that were found in both the soil and SbRB communities (as shown in Supplementary Table 3), of which a *Flavobacterium* ASV and a

Serratia ASV accounted for the 43.3% of the RhizCom, as stated in the Results (L114-L115), with the abundance of the rest being <0.5% in the RhizCom (Supplementary Table 3).

Reference 26 (Abdelfattah et al., 2023) summarizes the current knowledge of the transmission routes of seed bacteria. Although there are still many unknowns and speculations, seed endophytes are present in different seed compartments (embryo, endosperm, perisperm) from which they can colonize the roots and leaves of the developing seedling, probably depending on the specific plant anatomy and physiology. To the best of our knowledge, how this happens requires further investigation. Thus, soil-dwelling bacteria can also become part of the rhizosphere community and the endosphere by entering through damaged root zones, lateral root emergence sites, etc.

How many bacterial isolates were obtained to represent SbRB? How do these compare to the results from the amplicon sequencing? In terms of number and identity? I see some information in Supplementary figure 3 and 4 but if I get it correctly, these were from amplicon and metagenomic sequencing?

We did not report the obtention of SbRB isolates in the first version of the manuscript. If the reviewer is referring to SbRB metagenome-assembled genomes (MAGs), we recovered 16 medium to high quality MAGs that were primarily assigned to the SbRB community. In terms of taxonomy, these MAGs represent the most abundant taxa of the SbRB community, namely *Pantoea*, *Paenibacillus*, *Rhizobium*, *Clavibacter*, *Sanguibacter*, *Microbacterium*, *Priestia*, *Pseudomonas*, etc. (see Supplementary Table 8). A direct comparison between the V3-V4 16S rRNA amplicon sequences and the 16S rRNA genes of these MAGs is not really reliable. Annotated 16S rRNA genes from MAGs often result from the “collapse” of reads from multiple 16S rRNA gene copies during metagenome assembly. As a result, these sequences are likely to represent a consensus sequence, the most abundant 16S rRNA gene copy, or, given the degree of strain heterogeneity present in MAGs, a collapsed/consensus sequence from multiple closely related strains. In addition, 16S rRNA genes were not identified in most MAGs (see Supplementary Table 8, column “Tax.16S”), precluding direct comparison.

As noted by Reviewer #1, Supplementary Fig. 3 and 4 present analyses based solely on 16S rRNA amplicon sequencing data, specifically alpha diversity indices and differential abundance of ASVs between conditions. We have revised the figure captions to clarify that these analyses are based solely on 16S rRNA amplicon sequence variants. Specifically:

1. Supplementary Fig. 3 caption: We have added “*based on 16S rRNA amplicon sequence variants. Alpha diversity measures [...]*”.
2. Supplementary Fig. 4 caption: Added “*of 16S rRNA amplicon sequence variants*” instead of “ASV” for clarity.

In addition, we have included the results of a new analysis validating the niche facilitation of SbRB bacteria by disaccharide metabolism. The results are shown in Extended Data Fig. 5 and in the text by adding the following:

1. Results section, L213-L216: “*Saccharide utilization profiles of five isolated RhizCom members were characterized in monocultures and co-cultures. Helper SbRB strains Pantoea and Paenibacillus enabled partner strains to grow on disaccharides as sole carbon source, confirming the role of SbRB as resource specialists and niche facilitators for other rhizosphere bacteria (Extended Data Fig. 5)*”.
2. Discussion section, L297-L299: “*This study highlights the key role of SbRB in shaping early rhizobiome assembly, as demonstrated by their ability to facilitate the growth of other rhizosphere bacteria through disaccharide metabolism (Extended Data Fig. 5)*.”
3. Methods: We have added a new section at the end, named “Bacterial isolation and saccharide utilization profiling”, which describes the methods used for isolation of the

- five bacteria, taxonomic classification, monosaccharide and disaccharide utilization profiling in monocultures, and co-culturing of helper SbRB strains with others impaired in disaccharide utilization.
4. Raw c.f.u. counts from the experiment have been added to Supplementary Table 9.
 5. 16S rRNA sequences of the five bacterial isolates have been deposited at NCBI GenBank. The accession numbers have been indicated in the Data availability statement.
 6. GitHub has been updated to include the raw file of CFU counts and the code used for their analysis. The Zenodo repository has also been updated accordingly.

I also find authors to use a lot of abbreviation SM, SE, RW, RE, SbRB, RhizCom, NatComs, STB etc. SE and SM SW and RW are specific to experimental conditions and not widely recognizable. Writing "soil extract" and "soil matrix" "soil wash" and "rhizosphere wash" could be written in full to enhance clarity.

We agree with Reviewer #1 that the overuse of abbreviations in the methods makes the manuscript difficult to read. This was done to make the text more concise. We have followed the reviewer's suggestion and the abbreviations SE, SM, SW, and RW have been written in full throughout the manuscript and its Supplementary Information.

Minor comments:

The introduction could benefit from focusing on the

We are sorry, the sentence did not reach us in full, and we do not know what Reviewer #1 was trying to say. However, the other two reviewers commented on the Introduction, and we did our best to revise it in accordance with both. We hope that their concerns overlapped somewhat with those of Reviewer #1. We also tried to condense some parts to save space, and removed descriptions of examples of microbiome manipulation.

Some figures are data-dense and might benefit from simplification or annotations to highlight key findings.

We thank Reviewer #1 for this suggestion. After careful review of the figures, we have made changes to Figure 2 to improve readability. We agree that the figures are dense, but each has been carefully thought out to convey a key message while adhering to the journal's guidelines.

Ensure all supplementary materials referenced are accessible and clearly indexed.

We have carefully re-checked all of the supplementary material and its references in the main text as encouraged by Reviewer #1, to ensure that they are all cited correctly.

Address potential biases in microbiome assembly due to the artificial setup of microcosms compared to natural conditions.

We thank Reviewer #1 for this comment, which also agrees with those of Reviewer #2 and Reviewer #3. We have added the following sentence to the Discussion to address this concern: "*Our microcosms may introduce biases compared to natural conditions, such as the lack of soil heterogeneity and the use of surface-disinfected seeds, which in nature carry bacteria that may contribute to rhizobiome assembly*" (L259-L262).

Some terms like "niche facilitation" could be briefly defined for readers outside the field.

We thank Reviewer #1 for this comment. We have now briefly defined “niche facilitation” in the Discussion in the context of our work, in L257-L258 “[...] *niche facilitation, e.g., metabolic byproducts of one bacterium facilitating the growth and establishment of another*”. We are aware that the manuscript would be more accessible to a wider readership if more explanations and detailed descriptions were introduced. However, we are unable to expand the text further due to space limitations. We hope that the reviewer will understand these reasons.

Update references to include more recent studies where applicable especially on inheritance

We thank the reviewer for this suggestion. In order to maintain a concise number of references throughout the text, we believe that references 26 to 35 (of the revised version) already provide sufficient background on bacterial inheritance in plants and their roles. Additionally, we have included the following reference in the Discussion (L248, “*with their [SbRB] influence varying across different soil microbiome compositions*”), which examines seed inheritance across multiple soils, also using wheat as a model plant:

1. Walsh et al. Variable influences of soil and seed-associated bacterial communities on the assembly of seedling microbiomes, *The ISME Journal*. 2021;15(9):2748–2762.

As requested by Reviewer #3 we have also added the following two references to the Introduction (L68):

2. Matsumoto, H. et al. Bacterial seed endophyte shapes disease resistance in rice. *Nat. Plants* 7, 60–72 (2021).
3. Bergna, A. et al. Tomato seeds preferably transmit plant beneficial endophytes. *Phytobiomes J.* 2, 183–193 (2018).

We hope these additional references provide more background on bacterial inheritance in plants and its importance.

Reviewer #2 (Remarks to the Author):

Your paper, “Heritable seed-borne bacteria drive wheat rhizosphere microbiome assembly...” was very clear in its observations and conclusions. By using amplicon sequencing to follow bacterial communities over the course of several generations of rhizosphere transfer, you were able to show that soil inoculated wheat rhizospheres looked very much like the control rhizospheres which were grown without any external inoculum. For this section you have all the necessary details in the supplemental methods where few people will ever look, and one line buried in the methods, seemingly as an afterthought saying you had an uninoculated treatment. For most of the paper I was confused how the SbRB was identified or defined but I think it is critical to mention early on in the paper how and why you generated that “negative control” or SbRB. I think you should make this methodological detail clear in both the abstract and the introduction, clearly explaining that you had two different treatments in your experiment: serial transfer of rhizospheres from soil inoculated or uninoculated plants.

We are grateful to Reviewer #2 for dedicating the time to review our manuscript and for the insightful feedback which greatly helped us to refine and improve our manuscript.

Please note that all line number references in our responses correspond to the revised version of the manuscript. Given the strict text limitations imposed by the journal, we have addressed the reviewer’s concerns as concisely as possible, while making every effort to incorporate all suggestions by adjusting and streamlining other sections of the manuscript.

We appreciate the concern regarding the lack of clarity about how we obtained the SbRB community. Following Reviewer #2 comments, we have modified the manuscript to make this clear throughout the text. Specifically:

1. Results: We have added the following sentence: “*The SbRB community was obtained from one cycle of 16 uninoculated plants*”, L90.
2. Methods: We expanded the description of how we obtained the SbRB community, also in response to Reviewer #1, which now reads: “*To recover the seed-borne rhizosphere bacteria (SbRB) community, 16 microcosms were set up for a cycle (seven days) of plant growth without addition of soil inoculum. After seven days, the rhizospheres were collected, pooled by 4, and processed as described above*”, L346-L349.
3. Discussion: Also in agreement with Reviewer #1, we have added a sentence to highlight potential biases of our microcosms, including the use of surface disinfected seeds: “*Our microcosms may introduce biases compared to natural conditions, such as the lack of soil heterogeneity and the use of surface-disinfected seeds, which in nature carry bacteria that may contribute to rhizobiome assembly*”, L259-L262. This new paragraph also answers a few more questions from Reviewer #2. See below.

We hope this clarifies the obtention of the SbRB community. We prefer not to clarify it further in the Abstract or Introduction, given the very limited number of words allowed. We also apologize for having most of the details in the Supplementary methods, but we wanted to maintain a relatively reasonable text extension in the main text.

Also very important to mention both in the introduction and methods (but not necessarily the abstract) is that you grew the plant on a heat sterilized substrate and that all the plants came from surface sterilized seed, where you eliminated the influence of any seed surface microbiomes. One of the main reasons scientists have for so long thought that the plant microbiome derives from the soil, is that they never include negative controls (sterile soil and sterile inoculum) in their experiments and you should make clearer that you didn't continue making that mistake.

We agree with Reviewer #2 and appreciate these comments. As noted above, we have added a paragraph in the Discussion that addresses both the potential biases of the sterile soil matrix and surface disinfected seeds used in our microcosms. These discussion lines are in L259-L262, as noted above. In addition, we have added the sentence “Plants were grown from surface-disinfected seeds in microcosms with sterilized soil matrix” (L74-L75) in the Introduction. We have also clarified this in the Methods, including the use of a “*sterilized soil matrix*” and the use of “*surface-disinfected seeds*” (L324-L325).

We agree with Reviewer #2 that the lack of such controls has prevented scientists in the past from recognizing the importance of seed-transmitted bacteria. Many continue to do so. We hope that our work will further highlight this issue and the need to pay more attention to seed bacteria in any experimental design involving plants.

The metagenomics section was extensive, thorough and convincing in support of your hypothesis that SbRB are able to dominate the wheat rhizosphere thanks to their abilities to degrade specific root secreted saccharides. My only comment here is to ask if you could include a specific look at bacterial functions which may be beneficial to plants regardless of their enrichment, for example genes involved in nitrogen fixation, hormone production, siderophores, antibiotic production (ie. phenazines), etc. This might be a subfigure in your Figure 3, perhaps replacing 3C which seems unnecessary. Currently the only functions you seemed to look at were the most abundant/enriched, while it would be very interesting to many readers such as myself to know whether seed transmitted bacteria already bring with them most of the functions that wheat plants might desire to have in their rhizospheres.

We agree with the importance of this point raised by Reviewer #2. Although we initially considered including this in the manuscript, due to space limitations, we had to leave out other relevant and interesting analyses, such as the search for plant-beneficial functions,

and focus on the main messages of our research. We have tried to overcome this limitation by providing an Excel table in Supplementary Table 6 that lists all the functions categorized in the different “dominance categories”. Interested researchers could then filter this table to get the information they were curious about. For example, the first column “Dominance category” could be filtered to include all conditions, and then the table could be sorted by the column “SbRB_mean” (i.e., mean TPMs across the 3 replicates of SbRB). This can give an idea of all the functions present in the SbRB community. The column “GeneName” or “KEGGFUN” can be further filtered to search for specific functions. For example, filtering by the search term “phz” in the “GeneName” column, permits to see the genes *phzI*, *phzS*, *phzD*, *phzR* and *phzM*, of which only *phzI* and *phzS* are present in the SbRB (at very low abundance).

We believe that providing a list of plant beneficial genes that may or may not correspond to specific functions would also detract from the main message of the manuscript. We hope that Reviewer #2 will understand our reluctance to include such data analysis in the manuscript. We hope that Supplementary Table 6 can help provide a broader view of the functions that may be present in each community.

In addition, we have included the results of a new analysis validating the niche facilitation of SbRB bacteria by disaccharide metabolism. The results are shown in Extended Data Fig. 5 and in the text by adding the following:

1. Results section, L213-L216: “*Saccharide utilization profiles of five isolated RhizCom members were characterized in monocultures and co-cultures. Helper SbRB strains Pantoea and Paenibacillus enabled partner strains to grow on disaccharides as sole carbon source, confirming the role of SbRB as resource specialists and niche facilitators for other rhizosphere bacteria (Extended Data Fig. 5)*”.
2. Discussion section, L297-L299: “*This study highlights the key role of SbRB in shaping early rhizobiome assembly, as demonstrated by their ability to facilitate the growth of other rhizosphere bacteria through disaccharide metabolism (Extended Data Fig. 5)*.”
3. Methods: We have added a new section at the end, named “Bacterial isolation and saccharide utilization profiling”, which describes the methods used for isolation of the five bacteria, taxonomic classification, monosaccharide and disaccharide utilization profiling in monocultures, and co-culturing of helper SbRB strains with others impaired in disaccharide utilization.
4. Raw c.f.u. counts from the experiment have been added to Supplementary Table 9.
5. 16S rRNA sequences of the five bacterial isolates have been deposited at NCBI GenBank. The accession numbers have been indicated in the Data availability statement.
6. GitHub has been updated to include the raw file of CFU counts and the code used for their analysis. The Zenodo repository has also been updated accordingly.

In your discussion, you do a great job going over your results, but you haven’t done much to contextualize your experiment in relation to other plants or what is already known about microbiome transmission.

1. Were your wheat rhizobiomes similar to those observed in other published studies?
2. Do you think that wheat rhizobiomes form in the same way that all other plants do (ie. other monocots like maize or dicots like Arabidopsis and soybean)?
3. Why did you choose to work with wheat versus Arabidopsis for example?
4. How do seed transmitted bacteria exit the endosphere and colonize the rhizosphere?
5. Have you heard of the rhizophagy cycle and are most of these bacteria participants in that process?
6. Were any of the rhizobacteria you observed, part of the seed transmitted core microbiomes that are described in your references 28 and 29?

7. It may be useful to discuss whether there were any limitations to your study, for example, might there have been some sort of artifact in microbiome transmission created by sterilizing the seed surface before planting?

We appreciate these questions raised by Reviewer #2. For a clearer response, we have enumerated them as in the reviewer's comment. We agree with Reviewer #2 that this brings a very interesting discussion to our work that is worth exploring. Unfortunately, we have to focus on the most important questions of the manuscript, given the text length limitations. Nevertheless, we have reduced the text length elsewhere (Introduction and Results) to address these questions.

1. Yes, our RhizCom is similar to others previously observed in terms of major taxonomic groups and functions. This was already discussed in L276-L286 (of the revised version). While in this paragraph we refer to the functional differences of the communities, these functions are usually carried by the same bacteria (general taxonomy). We have added a new reference in the Discussion (Walsh et al., 2021, Ref 43) that further emphasizes that the inheritance of SbRB is variable across different soil microbiome compositions ("*with their [seed bacteria] influence varying across different soil microbiome compositions*", L248). Additionally, we have also added that the selection of taxa by the rhizosphere effect has been previously observed, in L238 of the Discussion, which now reads: "*This favored the assembly of Proteobacteria and Bacteroidetes on the wheat roots to the detriment of soil-dwelling Acidobacteria or Plantomycetes (Fig. 1e), as previously observed^{41,42}.*"
2. It is known that the composition of root exudates varies depending on plant species, cultivar, plant developmental stage, external factors, etc. In addition, plant secondary metabolites such as benzoxazinoids, coumarins, flavones, can greatly influence the composition of the rhizobiome. The profiles of these compounds are also highly variable among plant species/cultivars and influenced by external factors. We believe that it is plausible to assume that general ecological principles of microbiome assembly may be common to different plant species, but the specific mechanisms (i.e., taxa involved, resource utilization, etc.) may indeed be very different. We believe that this is indeed worth exploring. We have briefly pointed to this possibility in the Discussion, "*Future work should investigate whether these processes extend to other plant species and cultivars*", L299-L300".
3. The choice of the plant species and cultivar used in this study is based on the fact that it is one of the most important crops grown in Switzerland and available from the national seed bank. It is also the 3rd most productive crop in the world and the source of food for millions of people. To our best knowledge, *Arabidopsis* seeds do not transmit bacteria (probably due to decades of adaptation to laboratory conditions). While there are certainly more tools available for such a model plant (both plant genetics and associated microorganisms, i.e. the *At*-SPHERE collection), mechanisms identified in *Arabidopsis* may not be fully transferable to crop plants (or they may be! which would be worth exploring). We agree that studying how the seed bacteria of other plants and cultivars influence the rhizosphere microbiome is a very interesting topic that we plan to explore in future studies. These results would indeed allow a generalization of the mechanisms presented here or point to host-specific mechanisms.
4. Reference 26 (Abdelfattah et al., 2023) summarizes the current knowledge of the transmission routes of seed bacteria. Although there are still many unknowns and speculations, seed endophytes are present in different seed compartments (embryo, endosperm, perisperm) from which they can colonize the roots and leaves of the developing seedling, probably depending on the specific plant anatomy and physiology. To the best of our knowledge, how this happens requires further investigation.
5. We thank Reviewer #2 for pointing out the rhizophagy cycles. Indeed, the involvement of SbRB in rhizophagy cycles is a likely possibility, which could explain

how they are vertically transmitted. We recognize that further studies are needed. Our research alone is not sufficient to allow us to draw conclusions rather than speculate. Would these SbRB be found in the next generation of the plant? Will they be found as root endophytes? As aerial endophytes? These questions are certainly worth investigating.

6. Yes, the dominant SbRB are part of the core microbiota described in these references. Especially *Pantoea* and *Paenibacillus*. This, together with the fact that these bacteria are usually found as endophytes (Reference 32, Chen et al., 2017), further supports the idea that they are involved in rhizophagy cycles.
7. Thank you for this comment. In agreement with the other reviewers, we have added the following sentence to the Discussion: “*Our microcosms may introduce biases compared to natural conditions, such as the lack of soil heterogeneity and the use of surface-disinfected seeds, which in nature carry bacteria that may contribute to rhizobiome assembly*”, L259-L262.

We hope that our answers and changes are satisfactory. We appreciate these questions, which we will take into account for future lines of research.

Reviewer #3 (Remarks to the Author):

The authors suggest a sequential propagation strategy to generate a complex and reproducible plant rhizosphere community (RhizCom) to analyse plant microbiome assembly in wheat. Seed-borne rhizosphere bacteria emerged as the dominant microbiome source because of their ability to degrade specific saccharides and niche facilitation. The strength of the manuscript is the suggestion of the reproducible RhizCom to figure out the core of transmitted bacteria and their metabolic pattern (assimilation of disaccharides). However, I would suggest to consider the following comments and questions to improve the manuscript:

We thank Reviewer #3 for the time and effort invested in reviewing our manuscript and for the valuable feedback provided, which greatly helped us strengthen and improve our manuscript.

Please note that all line number references in our responses correspond to the revised version of the manuscript. Given the strict text limitations imposed by the journal, we have addressed the reviewer’s concerns as concisely as possible, while making every effort to incorporate all suggestions by adjusting and streamlining other sections of the manuscript.

Abstract

Please exactly justify the statement in what: Our results advance our understanding of the principles governing microbial community dynamics in early plant development.

We thank Reviewer #3 for this comment. We agree that the statement was vague. We have now rephrased it as follows: “*Our results advance our understanding of the importance of seed microbiota priority effects, niche preemption, partitioning, and facilitation in governing microbial succession and community assembly in early plant development [...]*”, L27-L30.

Unusual abbreviations, e.g. seed-borne rhizosphere bacteria (SbRB) seed-transmitted bacteria (STB) make it difficult to read. Are all these bacteria endophytes? – this is crucial to show.

We thank Reviewer #3 for these comments. Since the seeds were surface disinfected prior to the experiments (now emphasized throughout the manuscript: L75, L261, L325), all SbRB can be considered seed endophytes. However, we deliberately avoided using the term “endophytes” because we did not examine their location within the seed tissues and therefore referring to them as “endophytes” did not seem accurate enough.

We made a distinction between seed-transmitted bacteria (STB) and seed-borne rhizosphere bacteria (SbRB). STB broadly refers to bacteria that are transferred from the seed to any part of the new plant. In contrast, SbRB specifically describes the subset of STB that colonize the rhizosphere. This distinction is essential in our study to accurately describe the microbial ecology of the wheat rhizosphere. To improve readability, we have now rephrased sentences in L64-L72 for clarity and have omitted the abbreviation STB.

Add a few more details about the 821 metagenome-assembled genomes.

We thank Reviewer #3 for this suggestion. Due to text length restrictions imposed by the journal, we cannot devote more space to provide details of the 821 MAGs in the main text. We believe that Figure 4abc and Supplementary Table 8, provide sufficient information such as length, GC%, number of contigs, and statistics of completeness, contamination, strain heterogeneity, etc., per MAG. We hope this is sufficient.

Knowledge gap

We believe that after incorporating all of the suggestions made by the reviewers in the revised version, the knowledge gap has been further emphasized, especially in the Abstract (L27-L30) and Introduction (L69-L72).

Line 48: surprising, as soil and rhizosphere microbiomes are among the most diverse environments on Earth – this is true for the soil microbiome but not for the rhizosphere!

We thank Reviewer #3 for pointing this out. We have rephrased the sentence and changed “diverse” to “complex (L47), which applies to both soil and rhizosphere microbiomes.

Lines 48-57: There are not only these two approaches to get mechanistic insights! Here you can find examples for this:

Matsumoto H, Fan X, Wang Y, Kusstatscher P, Duan J, Wu S, Chen S, Qiao K, Wang Y, Ma B, Zhu G, Hashidoko Y, Berg G, Cernava T, Wang M. Bacterial seed endophyte shapes disease resistance in rice. *Nat Plants*. 2021 Jan;7(1):60-72.

A Bergna, T Cernava, M Rändler, R Grosch, C Zachow, G Berg. Tomato seeds preferably transmit plant beneficial endophytes. *Phytobiomes Journal* 2 (4), 183-193

We thank Reviewer #3 for this insight into other valuable approaches to gaining knowledge about mechanistic processes. We agree with their importance. We believe that introducing other approaches in this paragraph will disrupt the reading flow. However, we have included the two suggested references in L67-L68, where we believe they fit better as we discuss that seed-transmitted bacteria contribute to beneficial functions for the plant. “*These bacteria [SbRB] contribute to beneficial functions for the plant (REFs)*”.

Line 64: emphasized that seed-transmitted bacteria (STB) are also an important source of the rhizobiome.

We thank Reviewer #3 for pointing this out. We have reworded the paragraph to avoid the use of STB, as it does not appear elsewhere in the text (L64-L72). We hope that this change, along with the others throughout the manuscript, has improved the overall contextualization of seed bacteria. We apologize for not being able to expand the text further given the strict length restrictions imposed by the journal. We hope that the Introduction and Discussion fully emphasize the importance of seed bacteria.

Line 70: However, how both SbRB and soil microbiomes coalesce into the plant rhizobiome and whether specific metabolic signatures exist within the heritable plant microbial diversity

remains poorly investigated.

This paragraph needs more details – here, you should summarize current results and identify a clear knowledge gap. Mention research hypothesis would help as well.

We thank Reviewer #3 for this comment. In accordance with one of the previous comments, we have modified the sentence as follows: “However, *the mechanisms by which SbRB and soil microbiomes coalesce into the plant rhizobiome, and whether specific metabolic signatures drive their integration and function remain poorly investigated, hindering our ability to predict or manipulate these interactions*”, L69-L72. We apologize for not being able to fully address this suggestion. The only way to accommodate a longer introduction would be to reduce it from the Results or Discussion sections. We believe that these are already sufficiently condensed. We hope that Reviewer #3 will understand these reasons and that the added change will provide more clarity to the research hypothesis of our work.

Experimental design

The study focusses on bacteria only but also fungi are transmitted by seeds and wheat is ubiquitously mycorrhizal.

We agree with Reviewer #3 and acknowledge that fungi are indeed an important component of the rhizosphere microbiome, also transmitted by seeds. However, we believe that the focus on bacteria is sufficiently clear throughout the paper. In addition, the successive cycles probably prevented the establishment of fungi in the RhizCom (as observed by the low abundance of reads belonging to fungi, Supplementary Figure 5a), probably due to a slower growth rate.

Rhizobiome was frequently used – how is it defined? Ref?

It was defined early in the Introduction section, L40: “*the rhizosphere microbiome (i.e., rhizobiome)*”. We went through the manuscript and changed the remaining “rhizosphere microbiome” to “rhizobiome” after the first definition. This also helped to reduce the word count of the document. Thank you.

Plant microbiome assembly strongly depends on the species and even on the cultivar. What was the reason to select this one? To include more species or cultivars would allow more general answers and show that Fig 6 summarize a general picture. See Ref: Michl K, Berg G, Cernava T. The microbiome of cereal plants: The current state of knowledge and the potential for future applications. *Environ Microbiome*. 2023 Mar 31;18(1):28.

We thank Reviewer #3 for pointing this out. We agree. The choice of cultivar is based on the fact that Arina is one of the main wheat cultivars grown throughout Switzerland and available from the national seed bank. We agree that studying how the seed bacteria of other cultivars and plant species influence the rhizosphere microbiome is a very interesting topic that we plan to explore in future studies. These results would indeed allow a generalization of the mechanisms presented here or point to host-specific mechanisms. We have added this to the Discussion “*Future work should investigate whether these processes extend to other plant species and cultivars*”, L299-L300.

In addition, following up on this question, we have included in the Discussion (L238) that the general patterns of taxa selection in the rhizosphere have been observed previously: “*This favored the assembly of Proteobacteria and Bacteroidetes on the wheat roots to the detriment of soil-dwelling Acidobacteria or Plantomycetes (Fig. 1e), as previously observed^{41,42}.*”

Where the „successive cycles of microbiome propagation“ published before or is it your development? This is important for novelty of the study.

We thank Reviewer #3 for pointing this out. All data and experiments reported in the manuscript, including the successive cycles of microbiome propagation, are original and have not been published or submitted for publication elsewhere. We have modified the last paragraph of the Introduction and changed “*we used a sequential...*” by “*we developed a sequential...*” to emphasize the originality of our data.

Results

Line 157: the soil was enriched in functions – soil is more a seed and storage bank for bacteria, I don't think that enrich is the right word here

We appreciate this comment of Reviewer #3 and agree that the soil itself does not actively become enriched in functions. In this context, the term “enrich” is used to describe the result of the differential abundance analysis, which indicates functions that are statistically more abundant in the soil community compared to the other two communities. This analytical framework allows us to identify functions that are relatively overrepresented within each community. While we understand that “enrich” may suggest an active process, we believe it is appropriate within the specific context of our analysis and the paragraph describing the results.

Line 203-204: nicotinate, and nicotinamide metabolism and riboflavin metabolism – what does this mean?

We thank Reviewer #3 for pointing this out. The terms “nicotinate and nicotinamide metabolism” and “riboflavin metabolism” refer to functional categories defined in the KEGG database that include genes involved in the biosynthesis, conversion, or degradation of these vitamins. Our results indicate that genes within these categories are significantly more abundant in the SbRB community. As discussed later in the manuscript, we further identified complete pathways for the biosynthesis of both vitamins, predominantly in SbRB MAGs, whereas these pathways were largely absent in soil MAGs (Figure 5).

Line 2021: hat SbRB initiate the assimilation of disaccharides in the wheat rhizosphere, which can then be utilized by a larger number of RhizCom members.
I mean this does not explain the selective enrichment in the rhizosphere because all bacteria would love that?

Our data show that only certain SbRB have the metabolic potential to utilize disaccharides. The byproducts of this metabolism, namely glucose and fructose, could indeed be utilized by many rhizosphere bacteria, as shown in Fig. 5. The selective enrichment in the rhizosphere, as stated throughout the manuscript, is due to priority effects, niche preemption and niche partitioning by disaccharide specialization. In this sense, SbRB, as the first to have access to both disaccharides and monosaccharides exuded by plant roots, would have initially focused on the utilization of monosaccharides because their utilization is metabolically less costly (i.e., disaccharides require additional degradation steps). This leads to niche preemption. Thus, even if other bacteria could certainly use monosaccharides to grow, these resources are already used and preempted by SbRB, limiting the establishment of other bacteria. Even in the case that some soil bacteria could outcompete SbRB (i.e., antibiotics, weapons, etc.), the fact that SbRB are the only ones able to utilize disaccharides would have particularly allowed them to establish and persist on the roots even when there is competition for monosaccharides (for example during later root colonization).

In addition, we have included the results of a new analysis validating the niche facilitation of SbRB bacteria by disaccharide metabolism. The results are shown in Extended Data Fig. 5 and in the text by adding the following:

1. Results section, L213-L216: “*Saccharide utilization profiles of five isolated RhizCom members were characterized in monocultures and co-cultures. Helper SbRB strains Pantoea and Paenibacillus enabled partner strains to grow on disaccharides as sole carbon source, confirming the role of SbRB as resource specialists and niche facilitators for other rhizosphere bacteria (Extended Data Fig. 5)*”.
2. Discussion section, L2975-L299: “*This study highlights the key role of SbRB in shaping early rhizobiome assembly, as demonstrated by their ability to facilitate the growth of other rhizosphere bacteria through disaccharide metabolism (Extended Data Fig. 5)*.”
3. Methods: We have added a new section at the end, named “Bacterial isolation and saccharide utilization profiling”, which describes the methods used for isolation of the five bacteria, taxonomic classification, monosaccharide and disaccharide utilization profiling in monocultures, and co-culturing of helper SbRB strains with others impaired in disaccharide utilization.
4. Raw c.f.u. counts from the experiment have been added to Supplementary Table 9.
5. 16S rRNA sequences of the five bacterial isolates have been deposited at NCBI GenBank. The accession numbers have been indicated in the Data availability statement.
6. GitHub has been updated to include the raw file of CFU counts and the code used for their analysis. The Zenodo repository has also been updated accordingly.

Please show the overlap of amplicons and MAGs in more detail.

A direct comparison between the V3-V4 16S rRNA amplicon sequences and the 16S rRNA genes of these MAGs is not really reliable. Annotated 16S rRNA genes from MAGs often result from the “collapse” of reads from multiple 16S rRNA gene copies during metagenome assembly. As a result, these sequences are likely to represent a consensus sequence, the most abundant 16S rRNA gene copy, or, given the degree of strain heterogeneity present in MAGs, a collapsed/consensus sequence from multiple closely related strains. In addition, 16S rRNA genes were not identified in most MAGs (see Supplementary Table 8, column “Tax.16S”), precluding direct comparison.

Fig 6 try to integrate more details about the bacterial structure and function

We thank Reviewer #3 for this suggestion. We have included the most important SbRB taxa, *Pantoea*, *Paenibacillus* and *Priestia* in Fig. 6, specifically in the metabolism of disaccharides to monosaccharides and in the biosynthesis of vitamins. We believe that our results really highlight the role of these specific seed bacteria. We hope that this change has improved the figure. However, we do not feel confident enough to include other bacterial names or community structure as this may change along the root axis. We also believe that the key functions identified in this work are already included (i.e., disaccharide and monosaccharide metabolism).

Reviewer #3 (Remarks on code availability):

ok

Point-by-point response to reviewers

Version: 1

Reviewer #1 (Remarks to the Author):

Thank you for your detailed responses and revisions. Overall, I am pleased with the improvements made to the manuscript. I appreciate your thoughtful consideration of my comments and the effort to address them.

Best regards,
Reviewer #1

We again thank Reviewer #1 for the thoughtful and insightful comments, which greatly helped to improve the manuscript.

Reviewer #2 (Remarks to the Author):

Thank you for attempting to address my questions and comments. My most important requests were addressed, while a few less important ones were not possible due to length limitations. It is difficult to include so much information in a short, condensed publication and I applaud you for doing so. Try look a little more at rhizophagy in the future though. ;-)

We thank Reviewer #2 once again for the insightful comments on the manuscript. We appreciate your suggestion and will certainly explore the involvement of taxa in rhizophagy cycles in future studies. Thank you!

Reviewer #3 (Remarks to the Author):

The revision is really well done, and I am fully satisfied with the current manuscript. Congratulations!

We thank Reviewer #3 for the time and effort invested in reviewing our revised manuscript, as well as for the final constructive comments.

Only three minor things:

Line 59: typing error, remove one comma

Thank you for noticing! It has been corrected.

Line 63-64: "major source" for crops I would agree but for native plants, according to the current knowledge, the proportion of bacteria originating from seed or soil strongly depends on plant species. So may be rephrase a bit.

We agree with Reviewer #3. We wanted to emphasize a historical point where soil bacteria were considered to be the predominant source of rhizosphere bacteria until not so long ago. We agree that the study of rhizosphere bacteria has expanded greatly in recent decades, and other sources are now recognized, with indeed a variable influence among plant species. We have reworded the paragraph to emphasize this historical perspective and it now reads: "*Owing to the diffusion of root exudates into the adjacent soil, soil-dwelling bacteria have long been considered the predominant source of rhizosphere microbes. However, recent studies have emphasized that seed-transmitted bacteria are also an important source of the rhizobiome*". L61-L64

And only one question of personal interest: Pantoea is more and more considered as

opportunistic plant and human pathogens. F. e. EFSA has forbidden any application as plant growth promoting bacteria. Could you find any pathogenicity factor in your strains/MAGs?

Thank you for bringing this up for discussion. We are aware that *Pantoea* is considered an opportunistic human pathogen (as are many other plant-associated bacteria such as *Pseudomonas* or *Serratia*), although we were not previously aware of the EFSA regulation restricting its use. Similar to other genera that can colonize different environments, we believe that the potential risks associated with *Pantoea* are indeed largely dependent on the presence of specific pathogenicity factors and the host and environment dependent regulation of their expression, which may vary between strains and species.

We have examined the MAGs M057 and M058 assigned to *Pantoea* and, to the best of our knowledge, have not identified any known pathogenicity factors in their genome annotation, except for toxin-antitoxin systems (CptAB, ChpBS, HigAB, ParED), hemolysins, and a putative RTX toxin. Both MAGs harbor T6SS and T3SS systems as well as the siderophore desferrioxamine E. However, we acknowledge the possibility that some pathogenicity-related elements may be present but poorly annotated.

We are currently in the process of sequencing the whole genome of two *Pantoea* strains isolated from the RhizCom/SbRB and will pay special attention to this aspect in our analyses and future research. We are grateful to Reviewer #3 for bringing this to our attention.